

# Simulation of a former ice field with PISM – Snežnik study case

Matjaž Depolli[1], Manja Žebre[2], Uroš Stepišnik[3], and Gregor Kosec[1]

[1]Department of Communication Systems, Jožef Stefan Institute, Jamova cesta 39, SI-1000 Ljubljana, Slovenija
[2]Geological Survey of Slovenia, Dimičeva ulica 14, SI-1000, Ljubljana, Slovenia
[3]Faculty of Arts, University of Ljubljana, Aškerčeva cesta 2, SI-1000, Ljubljana, Slovenia

**Correspondence:** Matjaž Depolli (matjaz.depolli@ijs.si)

**Abstract.** In this paper we present a reconstruction of climate conditions during the Last Glacial Maximum on a karst plateau in Dinaric Mountains (southern Slovenia) that bares evidence of glaciation. The reconstruction merges geomorphological ice limits, classified as either clear or unclear, and computer modelling approach based on Parallel Ice Sheet Model, which is an established numerical model for simulating glacier dynamics ranging from ice sheets to alpine glaciers. Based on extensive numerical experiments, where we studied the agreements between simulated and geomorphological ice extent, we propose to use a combination of high resolution precipitation model that accounts for orographic precipitation combined with simple elevation based temperature model. The geomorphological ice extent can be simulated with climate around 6°C colder than modern and with a lower than modern amount of precipitation, which matches other state-of-the art climate reconstructions for the era. The results indicate that orographic precipitation model is essential for accurate simulation of the Snežnik with moist southern winds from the nearby Adriatic Sea having predominant effect on the precipitation patterns. Finally, this study shows that transforming climate conditions towards wetter and warmer or drier and colder does not significantly change conditions for glacier formation.

## 1 Introduction

The Last Glacial Maximum (LGM) in Europe was dominated by the Fennoscandinavian Ice Sheet, which was one of the major global ice masses during the last glacial cycle. A relatively large ice mass was also the Alpine Ice Sheet over the European Alps. Other, much smaller mountain glaciers existed in Iberia, the Pyrenees, the Apennines and the Balkans (Hughes et al., 2013). These smaller ice masses, being isolated from larger ice sheets, have a well-defined linkage between their areas of origin and the areas they affect, which simplifies the task of reconstructing their past extents and behaviors. Additionally, their tendency to react quickly to climate change makes them valuable proxies for deducing historical climate conditions.

Past glaciations in the northern Dinaric Mountains at the interface between the Alpine ice sheet and Balkan Mountain glaciers are well documented geomorphologically (e.g., Žebre et al. (2013), Žebre and Stepišnik (2016), but there is lack of knowledge about climate-glacier dynamics based on modelling approach. The northern Velebit Mountains in Croatia are the only formerly glaciated mountains in this region where empirical reconstruction has been compared with computer-based simulations under different palaeoclimate forcings (Žebre et al., 2021). In southern Slovenia only a few kilometers to the North of Velebit lies Snežnik, one of the northernmost mountain plateaux in the northern Dinaric Mountains. Snežnik was glaciated





during the LGM, and the maximum area covered by an ice field was estimated to be at least $40\,\mathrm{km}^2$ (Žebre and Stepišnik, 2016). Although the formerly glaciated area is very small compared to the large Alpine ice sheet (estimated to $163000\,\mathrm{km}^2$ by Seguinot et al. (2018)), detailed knowledge of it can still aid in deciphering the regional past climate conditions. There have been only a few attempts to model the palaeoice field on the Snežnik Mountain either only on the limited area around the small

Snežnik summit (Žebre and Stepišnik, 2016) or as part of a larger Alpine area (Seguinot et al., 2018). In the first case, a simple steady-state model that assumes a perfectly plastic ice rheology was applied (Benn and Hulton, 2010), while in the second case the Parallel Ice Sheet Model (PISM) was used (the PISM authors, 2023). However, modelling small palaeoglaciers like the one on Snežnik is challenging, especially due to the need for high-resolution climate data and proxy-based palaeoclimate forcings, but also due to insufficient knowledge of pre-ice topography and ice flow.

In this work, we focus on amending the geomorphological knowledge with a computer model focused on the whole Snežnik ice field. We aim to discover the climatological conditions required for the ice field to form in the extend that is evident from the field observations. We approach the modelling challenge by using the PISM on lidar-based topography with $50\,\mathrm{m}$ resolution and with standard glacier modelling approach (Bueler and Brown, 2009) and custom climate forcing models. Near-surface air temperature and precipitation are used as climate forcing inputs in PISM and we have developed custom models for both,

using the available topology, single weather station data and the knowledge of current wind patterns in the broader area. We present and explore several climate forcing models ranging from simple to complex. In case of Snežnik, the main challenge is achieving ice distribution according to the geomorphological evidence, which is skewed counter-intuitively, i.e., towards the well insolated southern slopes of the plateau. The inability of the simulations to achieve ice cover extent skewed similarly to the estimated extent can be attributed to inaccuracies in many of the used models, including sliding laws, basal conditions and

climate forcing. In this work, we chose to only address the latter, since a plethora of climate data is available that could lead to simulation improvement if integrated into the computer models.

## 2 Study area

Snežnik (45°34'53"N 14°25'53"E) is a wide karst plateau with an area of roughly $100\,\mathrm{km}^2$ in the northern part of Dinaric Mountains in Slovenia. The highest summit is Veliki Snežnik (1796 m asl).

### 2.1 Glacial geomorphology

Geomorphology of the area has been studied well with studies realizing the ice extent appearing as early as 1959 (Šifer, 1959). More recent works largely confirmed the previous findings (Žebre and Stepišnik, 2016), but also focused on detailed geomorphology, geochronology and glacial-karst interaction (Žebre et al., 2016, 2019). These found that most glacial deposits are present between 900 and 1200 m asl. They form characteristic glacial depositional features, which stand out from the

surrounding karstic area, whereas typical glacial erosional features are not common for the area. Instead, the area is dissected by glaciokarst depressions, which are most likely formed by a combination of karst processes and subglacial erosion (e.g., as described in Smart (1987); Žebre and Stepišnik (2016)). The maximum geomorphological ice extent in Snežnik was estimated



based on overall position of glacial features, which was subsequently divided into clear and unclear ice boundaries. The first were delineated based on end moraines and outwash fans, while the latter were drawn over areas that only show suggestive evidence for glaciation. The geochronological data (Marjanac et al., 2001; Žebre et al., 2019), although still scarce, points to a maximum ice extent during the last glacial maximum (LGM), that is 30–17 ka BP (Lambeck1 et al., 2014).

## 2.2 Climate

Plateau declines to the south and the southwest towards the Adriatic sea that starts around 25 km away. With the proximity of Adriatic sea, the plateau is provided with above average precipitation for the area. Snežnik receives between 2000 mm and 3000 mm of precipitation annually, with strong annual cycle (ARSO). There are two major drivers of precipitation, namely the Genoa low cyclone and the regular moist wind blowing from the south–southeast, which are intensified by the orographic topology. The mean annual air temperature at the meteorological station in Mašun (period 1971-1986), NW of Snežnik is around 7 °C, average air temperatures for winter and summer are -3 °C and 15 °C, respectively. The rise of the specified temperatures at Mašun, relative to the pre-industrial era, can be estimated from the average of Slovenia published by Urad za meteorologijo, hidrologijo in oceanografijo (2021), and is 0.4 °C relative to pre-industrial era. Global temperature at the LGM was 3 to 6 °C lower than in pre-industrial times (Annan and Hargreaves, 2013), whereas the LGM temperature offset over the Alps has been recently simulated to about -6.6 °C (Del Gobbo et al., 2023). As for the precipitation in the LGM, the same research suggests the Alps were overall drier (by ≈16%) than in pre-industrial times. Del Gobbo et al. (2023) found similar LGM temperature and precipitation offset for the Northern Dinaric Mountains in general, but allowing up to ≈30% drier climate around Snežnik. (Figure 3 in Del Gobbo et al. (2023)). During the last glacial cycle, sea level in the Adriatic was 100-130 m lower than present (Spratt and Lisiecki, 2016; Gowan et al., 2021), positioning Snežnik much further from the coastline and associated moisture source, at a distance of 150-200 km.

## 3 Methods

### 3.1 PISM setup

PISM 2.0 was installed on several Linux workstations and used throughout the experiments. The modelling parameters of PISM were set identically to those described in Žebre et al. (2021), where a larger mountain range just to the South of Snežnik was simulated, and similarly to those in (Candaş et al., 2020), where a similarly sized mountain in Turkey was simulated. That means that most of the parameters were actually left at their default values, which can be found in the official documentation (the PISM authors, 2023). The rest of parameters, e.i., those that were modified, and the inputs that were used are listed in the tables of this section. Model selection is listed first in Table 1. Next, the PISM parameters that either depend on the model selection or are general, are listed in Table 2. Finally the various settings that do not translate to PISM parameters directly are listed in Table 3.





**Table 1.** PISM models used in the study

| Model | Value | PISM option |
|---|---|---|
| Stress balance | SIA + SSA | -stress_balance ssa+sia |
| SIA flow law (rheology) | Glen-Paterson-Budd-Lliboutry-Duval law | -ssa_flow_law gpbld |
| Sliding law | pseudo-plastic power law model | -pseudo_plastic |
| Surface mass and energy process | Positive Degree Days (PDD) | -surface pdd |
| Atmospheric annual cycle | Scalar precipitation and temperature offset | -atmosphere given,delta_T,delta_P |
| Subglacial hydrology | Undrained plastic bed | -hydrology null |
| Ocean models | disabled | -dry |

**Table 2.** PISM parameters that do not depend on the domain size and resolution

| Parameter | Value | PISM option |
|---|---|---|
| Number of vertical layers in the ice | 21 | -Mz 21 |
| Calculation box height | 1000 m | -Lz 1000 |
| Number of vertical layers in the bedrock | 5 | -Mbz 5 |
| Bedrock calculation depth | 100 | -Lbz 100 |
| Prevailing wind direction | 150° | -atmosphere.orographic_precipitation.wind_direction 30 |
| Scaling factor for precipitation | 1 | -atmosphere.orographic_precipitation.scale_factor 1 |
| Latitude for calculating Coriolis effect | 45° | -precipitation atmosphere.orographic_precipitation.coriolis_latitude 45 |
| True background precipitation | 1946 $\mathrm{mm}/\mathrm{annum}$ | -atmosphere.orographic_precipitation.background_precip_post 1946 |
| Pre-applied background precipitation | 0 $\mathrm{mm}/\mathrm{annum}$ | -atmosphere.orographic_precipitation.background_precip_pre 0.0 |
| $q$ | 0 | -pseudo_plastic_q 0 |
| $u_{\mathrm{threshold}}$ | 100 $\mathrm{m}/\mathrm{annum}$ | -pseudo_plastic_uthreshold 100 |
| Till compressibility coefficient | 12 | -till_compressibility_coefficient 0.12 |
| Till reference void ratio | 1 | -till_reference_void_ratio 1 |
| Effective overburden pressure of till | 0.02 | -till_effective_fraction_overburden 0.02 |
| Tile reference effective pressure | 1000 | -till_reference_effective_pressure 1000 |
| Till water decay rate | 1 | -hydrology_tillwat_decay_rate 1 |
| Till water maximum level | 2 | -hydrology_tillwat_max 2 |
| $\phi_{\mathrm{plastic}}$ | 10 | -plastic_phi 10 |
| Till cohesion | 0 | -till_cohesion 0 |

**Table 3.** Parameters of custom models and other settings

| Parameter | Value |
|---|---|
| Total simulation length | 3000 years |
| Grid sequencing | 3 steps (see section 3.2 for details) |
| Insolation effect amplitude $A_S$ | 1 |



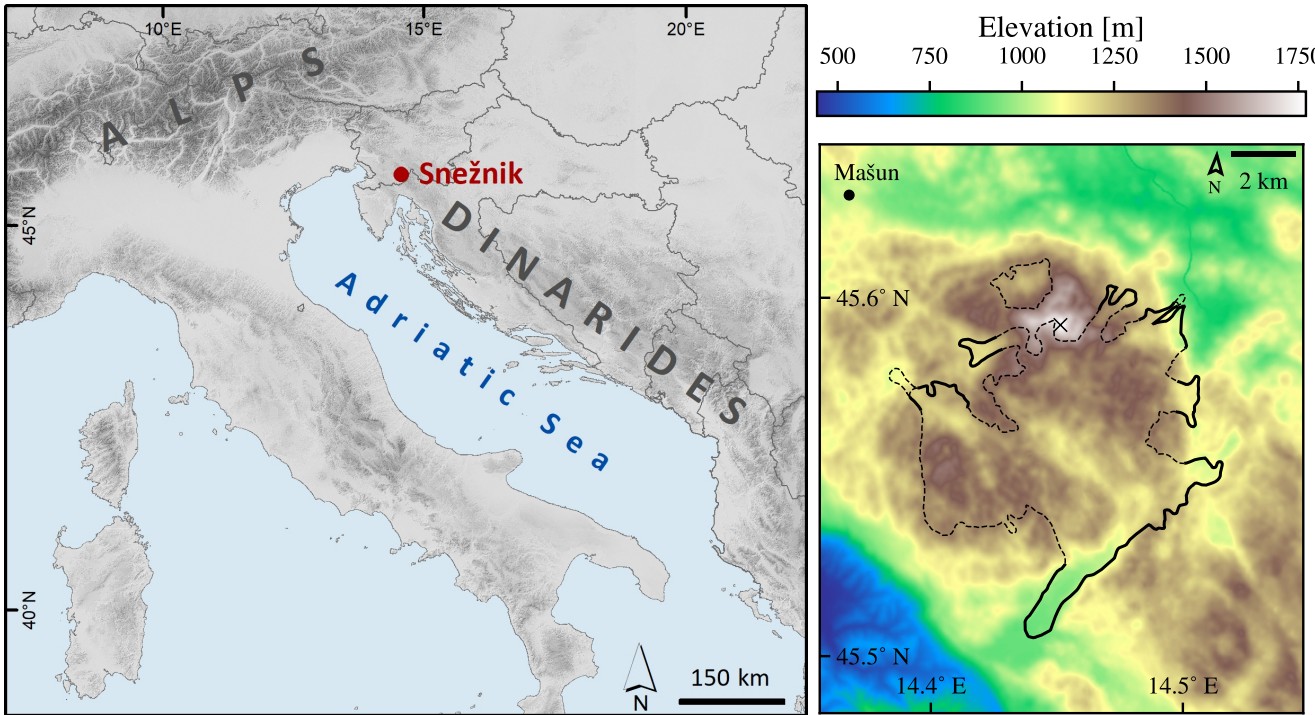

**Figure 1.** General location of the plateau Snežnik (left) and the domain in resolution of 25 m (right). The geomorphologically reconstructed glacier extent is plotted with dashed (unclear/estimated) and solid lines (clear/observed). Reference weather station and the highest summit of the plateau are marked with a dot and a cross, respectively.

## 3.2 Model domain

Source of the topographical data for the domain are the digital elevation maps (DEMs) provided by EU-DEM v1.1 in 50 m
resolution from (European Environmental Agency, 2016). To simulate the glacier, a rectangular domain is used that covers the area of about 266 km². For preliminary simulations, the resolutions of either 200 m or 150 m had been used, and most of the results presented in the paper have been computed with the resolution of 100 m and in a single case 50 m. Vertically, the ice thickness is limited to 1000 m, and the number of layers is set to 21, resulting in the vertical grid size of the computational box of 50 m. There are also 5 layers of simulated bedrock and the total height of simulated bedrock is 100 m. We experimented
also with higher vertical resolutions, but observed only a minuscule effect on the simulation results accompanied by a notice-able increase in execution time. For both, the number of ice and bedrock layers, we selected the values based on maximal performance with satisfactory level of details, which we determined in preliminary experimentation.

We used grid sequencing, an approach used to decrease the time complexity of steady-state simulations, which is supported by PISM. The simulation is started on a coarse grid for a large portion of simulation time or until a relevant metric, such
as the ice volume, converges. Then the grid is refined and simulation continues on the finer grid, i.e., the last state of the




simulation is interpolated to the finer grid (regridded) and taken as input into the next simulation stage. In the presented study we used the following setup for grid sequencing, which is based on the empirical values derived from preliminary experiments. Simulations were started on a 400 m grid for 500 years, then continued on 200 m grid for 1000 years, and finally on 100 m grid for 1500 years. Final resolution of these simulations was therefore 100 m, as is the resolution of results, while lower resolutions

are only used to form rough ice coverage from the initial no-ice conditions.

### 3.3    Quantitative validation

The main goal of the presented work is to create a computer model that can describe the steady-state ice field under LGM climate conditions. Since the general computer model for ice fields already exists but the climatic conditions on micro scale are only roughly known, this goal has been translated to a more immediate goal of determining the climatic conditions under

which the geomorphological shape of glacier can exist. This goal can be achieved through an optimization task – a set of input parameters which include the climate forcing has to be continuously tuned (optimized) for the simulations based on them to produce results of increasing quality. The quality of the results can be characterized by how closely the simulated glacier extent aligns with its observed geomorphological form. To provide for objective validation of simulation result quality, two quantitative metrics have been implemented.

First, a function to determine whether any given coordinate lies inside or outside of the geomorphological glacier area is constructed using both the clear (observed) and unclear glacier boundaries, which used together bound two separate geographical areas within the domain (see Figure 1). This function is used to divide the domain into geomorphological glacier area and the ice-free area. Then, only the clear glacier boundaries are used to generate two artificial border areas, one bounded by the observed boundary on one side and extending away from the ice field, and another bounded by the observed boundary on one

side and extending towards the interior of the ice field. These border areas thus represent an area where we are certain was ice free and an area where we are certain was covered with ice. The simulation should cover latter with ice but not the former. Therefore we name the two border areas *forbidden* and *required*, respectively.

The exact procedure to generate the two border areas is as follows. The clear limits are offset perpendicularly to itself in both directions (outwards and inwards), creating the boundaries of a singular *border area*. This is then compared to the

geomorphological glacier area, i.e., the area bordered by both clear and unclear geomorphologically reconstructed limits. The intersection of border area and the geomorphological glacier area then forms the required area, while their difference forms the forbidden area. Since both clear and unclear limits are used in the operation, it is assured that the forbidden area cannot extend into the geomorphological ice field area even in cases when offsetting the borders of clear limits does so, e.g. in nearby parallel outlet glaciers that exist in the northeast of the presented domain.

The optimization task for the simulated glacier should minimize the ice coverage of forbidden area and maximize the ice coverage of required area. While covered surface areas for border areas could be used in optimization procedure directly, we first transform them to relative metrics that are both to be maximized. Therefore, we define two performance metrics, sensitivity



Sen and specificity Spe, based on surface areas of the two border areas $A_{\text{required}}$ and $A_{\text{forbidden}}$:

$$\text{Sen} \quad = \quad \frac{A_{\text{required,covered}}}{A_{\text{required}}}$$

$$\text{Spe} \quad = \quad 1 - \frac{A_{\text{forbidden,covered}}}{A_{\text{forbidden}}}$$

When performing the validation of results, sensitivity and specificity are calculated and represent two objectives to be maximized. To illustration how the metrics are related to the results, we define a simple model of glaciation, where the existence of ice is a function of elevation. Areas above threshold elevation $h$ are covered with ice while the others are not, forming an ice field that is highly nonconforming with the geomorphological ice field shape. Relation of metrics to the thus defined glaciation

are shown Figure 2.

The amount by which borders are extended inwards and outwards to generate forbidden and required areas are parameters that we shall name *width* of the respective area. For simplicity we shall keep the widths of both areas equal. The presented metrics are a function of width, and thus the metrics can be tuned by adjusting the width as shown in columns of Figure 2. As the figure illustrates, a nonconforming ice field shape covers almost indiscriminately both the required and forbidden areas.

By increasing the size of a nonconforming shape, both border areas get covered more, therefore sensitivity increases but specificity decreases. Vice versa is true when the nonconforming shape decreases in size. To truly optimize the shape, one has to find ways to increase both sensitivity and specificity, which is only possible by transforming the simulated ice field into a more conforming shape.

An optimization where there are multiple criteria (or objectives) to be optimized, and these criteria are generally conflicting,

is called a multi-objective optimization (Collette and Siarry, 2004). The problem at hand is a multi-objective problem with two conflicting objectives, and must be solved as such. We use the "naive" approach (Collette and Siarry, 2004) to solving multi-objective problem and combine the two objectives into one: we construct the single objective Obj by multiplying sensitivity and specificity:

$$\text{Obj} = \text{Sen} \cdot \text{Spe} \tag{1}$$

Then we solve the simpler single-objective problem by searching for the parameters of simulation that maximize the value of the single objective. Such a combined objective is near zero when one of the factors is near zero, is higher for balanced than imbalanced factors (even if their sum is the same) and increases with the increase of each factor. As such, Obj serves well for optimizing simulated ice field shape to the prescribed form.

So far we have not picked the value for the width – the parameter of both border areas that was shown to greatly influence

the resulting sensitivity and specificity. We attempt to optimize the value for width in the following way. To use the presented metrics for quantitative result validation, the width should be large enough to cover the majority of simulated ice field borders and thus maximize the ability to assign different numerical value to different results. E.g. consider the setting in which the forbidden area extends only 1 km away from the clear boundary. Then, if a simulated glacier extends 2 km over the boundary, it will cause the same drop in the objective function as if it extended only 1 km over the boundary; the results of clearly different

quality will not be discernible through the proposed metrics. Thus large values of width should be preferred. On the other hand,



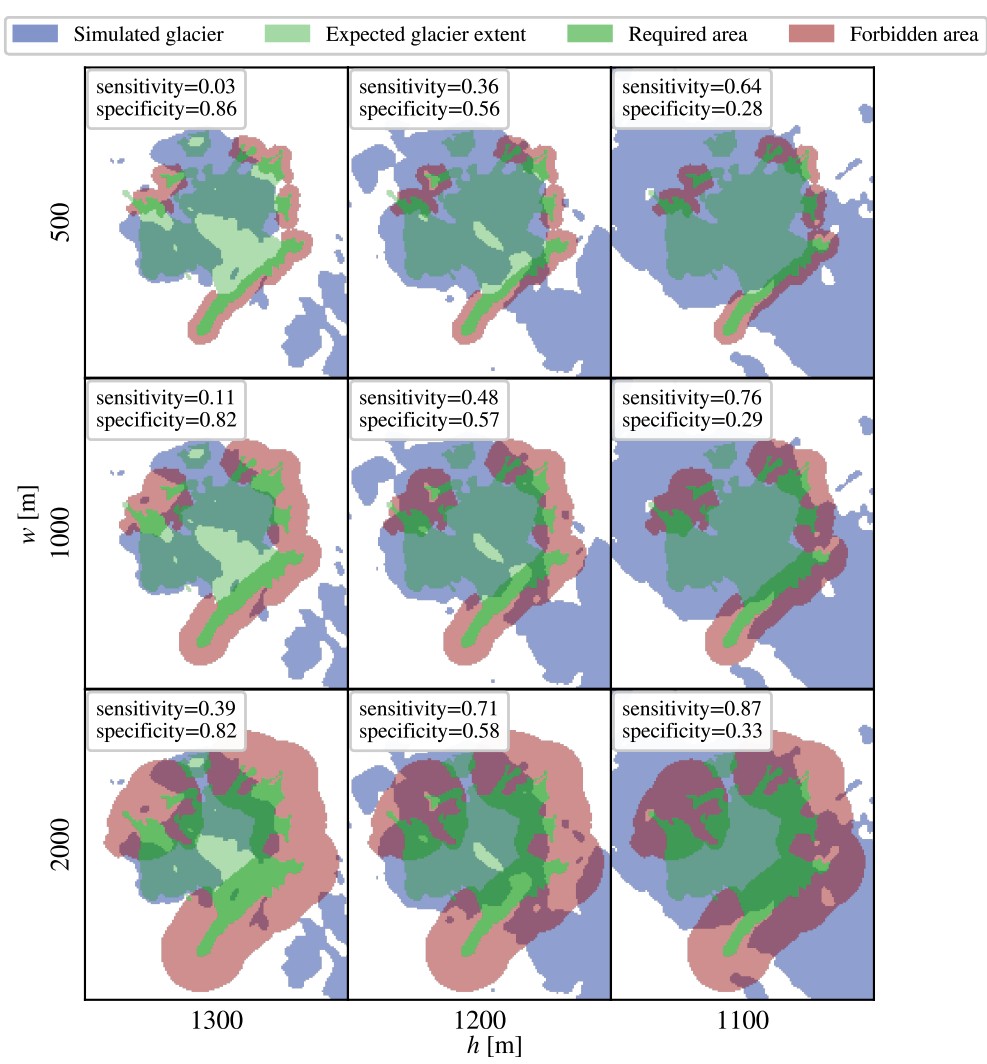

**Figure 2.** Illustration of the behaviour of quantitative performance metrics, demonstrated on three examples of ice coverage simulated by a simple model, where ice forms above the threshold elevation $h$ but not below. Metrics are calculated for varying width of forbidden and required areas $w$ (parameter of the metric), and varying threshold elevation $h$ (parameter of the simulation).





since the geomorphologically bound ice field is irregularly shaped, increasing the width will result in border areas spreading in such a way that the unclear limits will play increasingly large part in bounding them. This is not desired; the border area near unclear limits should be minimized to minimize the effect of errors that are likely present in unclear limits.

Three values for the width $w$ were displayed in Figure 2 for an elevation-based glaciation model and more were tested in preliminary experiments. The optimal width of border areas takes into account both the preference for large values and the desire to minimize the effect of unclear limits, and according to the scale of the major glacial features on the target area, seems to be $\approx 1000$ m. From the same figure, trends in sensitivity and specificity as functions of overall ice field extent (characterized by simulation parameter $h$) can be observed. For all tested values of $w$, the trends are similar, indicating that the exact value of $w$ might not be important in the context of optimization. Optimization works by analyzing the difference between objective values of several simulation results, not on their absolute values. Therefore, optimization is robust relative to the selection of $w$ and we do not try to fine tune $w$ any further.

## 3.4 Climate forcing

Climate forcing is implemented through temperature and precipitation models. Both provide location dependent monthly mean values that are kept constant throughout all the simulated years. Several different models were tested to gain insight into which aspects of climate influence the glacier formation the most and to see how much detail is required for simulations with high fidelity. In this section, two temperature and three precipitation models are presented.

The climate forcing models are initially tuned to match the modern conditions, for which a reference weather station is chosen. This weather station is located in a nearby Mašun, which lies within the area of plateau Snežnik, but outside of the geomorphological area of glaciation (45°37'41" N 14°21'59" E), on 1025 m asl. The station was active between years 1971 and 1986 and data is available for 97% of this time range. Monthly mean temperature and precipitation for the station are shown in Figure 3.

In the second step, the models are adjusted to shift the climate towards the LGM. The temperature model output is offset to lower its output by several degrees while the precipitation model output is multiplied by a factor to either increase or decrease the precipitation linearly by several percent to several ten percent. The adjustment to temperature and precipitation models are uniform across the domain, i.e. both the temperature offset and precipitation factor are not functions of location. The temperature offset and precipitation factor that form conditions for the optimal ice field are important results of the presented study.

### 3.4.1 Temperature

The baseline temperature model used in this study – we shall name it *lapse-rate model* – comprises a reference point from the reference weather station and the lapse rate for standard atmosphere (Atmosphere, 1975) (6.5 °C per km), from which temperatures for all grid points can be calculated:

$$T(x,y) = \frac{(h(x,y) - h_{\mathrm{ref}})}{1000 \cdot 6.5} + T_{\mathrm{ref}},$$



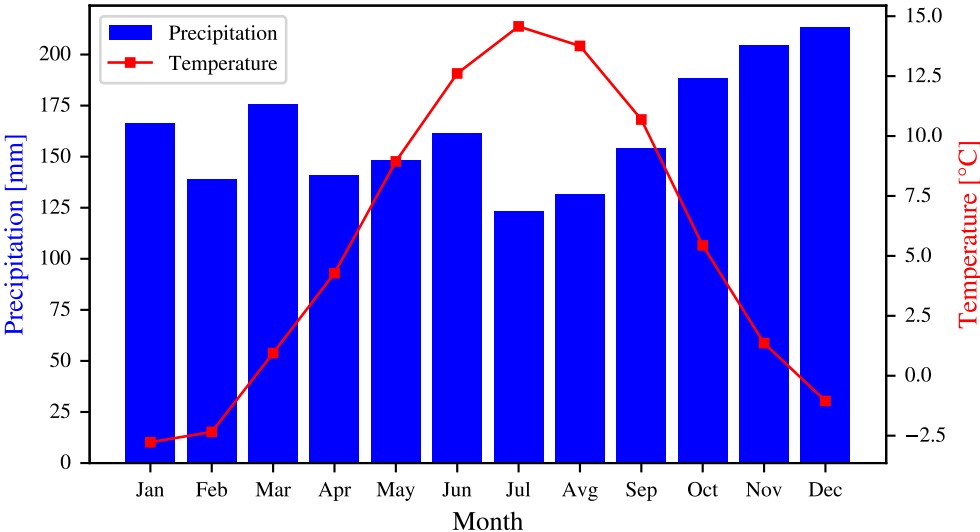

**Figure 3.** Mašun climate data.

where $h(x,y)$ is the elevation of each point in DEM, $h_{\text{ref}}$ is the elevation of the reference weather station and $T_{\text{ref}}$ is the mean temperature recorded at the reference weather station.

The second is *insolation adjusted lapse-rate model*, which extends the lapse-rate model with a proxy for the thermal radiation received from the Sun per surface area of the Earth (insolation). The insolation extension is a simplified version of the topographic shading model introduced by Olson and Rupper (2019). It calculates the relative proportion of Solar insolation hitting each grid element by projecting the sun radiance vector to the grid element normal, which is calculated numerically as a pair of symmetric differences through x and y axis. Therefore, the slopes perpendicular to the incoming radiation from the

Sun receive full relative insolation (value of 1), while slopes parallel to it receive no relative insolation (value of 0). The model of relative insolation $S$ is represented by the following equations (Burrough et al., 2015):

$$\begin{bmatrix} \frac{\partial h}{\partial x} \\ \frac{\partial h}{\partial y} \end{bmatrix} = \nabla h(x,y), \forall (x,y) \in \text{DEM}$$

$$\alpha = \frac{\pi}{2} - \arctan\left(\sqrt{\frac{\partial h}{\partial x}^2 + \frac{\partial h}{\partial y}^2}\right)$$

$$\beta = \frac{\pi}{2} - \arctan\left(\frac{-\frac{\partial h}{\partial x}}{\frac{\partial h}{\partial y}}\right)$$

$$S = \sin(\psi) \cdot \sin(\alpha) + \cos(\psi) \cdot \cos(\alpha) \cdot \sin(\beta)$$

where $\psi$ is the angle of the sun above the horizon, $\alpha$ is the angle of slope, relative to the horizon, and $beta$ is the aspect angle, i.e. the angle between south and the direction facing the steepest descend from the given coordinates.



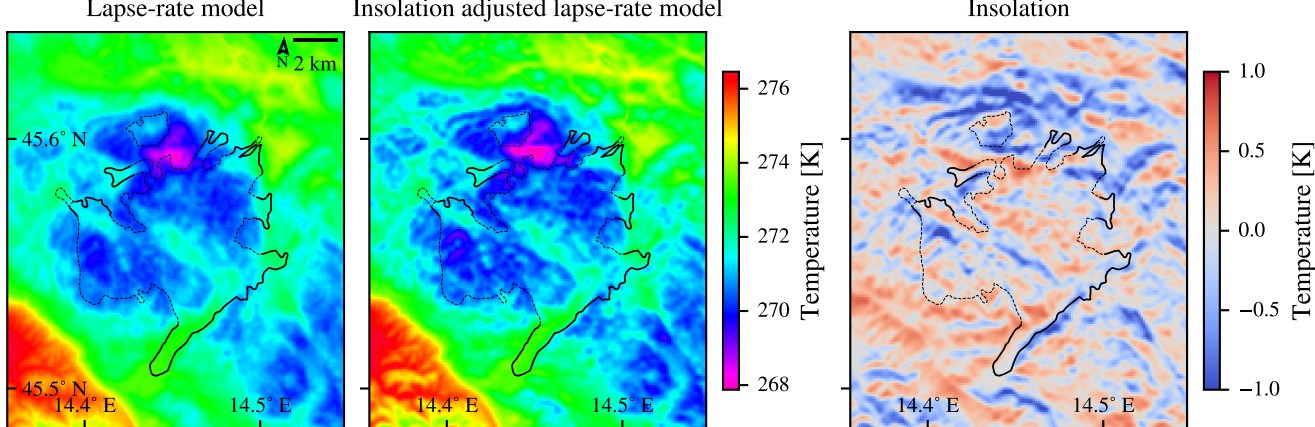

**Figure 4.** Comparison of the air surface temperature models. The generated temperature fields are on the left side while the insolation term of the insolation-adjusted lapse-rate temperature model (which equals the difference between the model outputs) is on the right side. The geomorphological glacial limits are plotted as black lines.

Relative insolation $S$ is then multiplied by *insolation effect amplitude* $A_S$ to form the equivalent difference in surface air temperature $T_S$:

$$T_S = A_S S.$$

The argument for such a mapping is that a fully insolated area is equivalent to a fully shaded area that is exposed to a higher surface air temperature. This is reflected in the Equilibrium Line Altitude (ELA), i.e., a boundary between the accumulation and ablation areas of the glacier, that is very sensitive not only to avalanching, snow drifting, glacier geometry and debris-cover but also to shading (Nesje, 2014). Geomorphological studies in a nearby Trnovski gozd (Kodelja et al., 2013) and (Žebre et al.,

2013) found out that difference between the ELA of the sun facing southern slopes and the shady northern slopes was at least 150 m while (Evans and Cox, 2005) sets the theoretical boundary of ELA difference at 320 m. The latter limit is used to derive the maximal insolation effect amplitude from the standard atmosphere lapse rate:

$$A_{S,\mathrm{max}} \approx \frac{320\,\mathrm{m}}{1000\,\mathrm{m}} 6.5\,\mathrm{K} = 2.08\,\mathrm{K}$$

Note that the calculation of $A_{S,\mathrm{max}}$ should be taken as a rough estimate.

For our further experiments we used a smaller value of $A_S = 1\,\mathrm{K}$, which matches the observed ELA difference better than the upper theoretical boundary. The comparison of temperature fields generated by the models is visualized in Figure 4. The difference (the insolation part of the model) can be seen to be small and local.

Finally, temperatures produced by any of the temperature models are offset (decreased) by a constant value, since the present climate is warmer than the simulated past. The preliminary experiments were used to tune the temperature offset to -5.6°C from



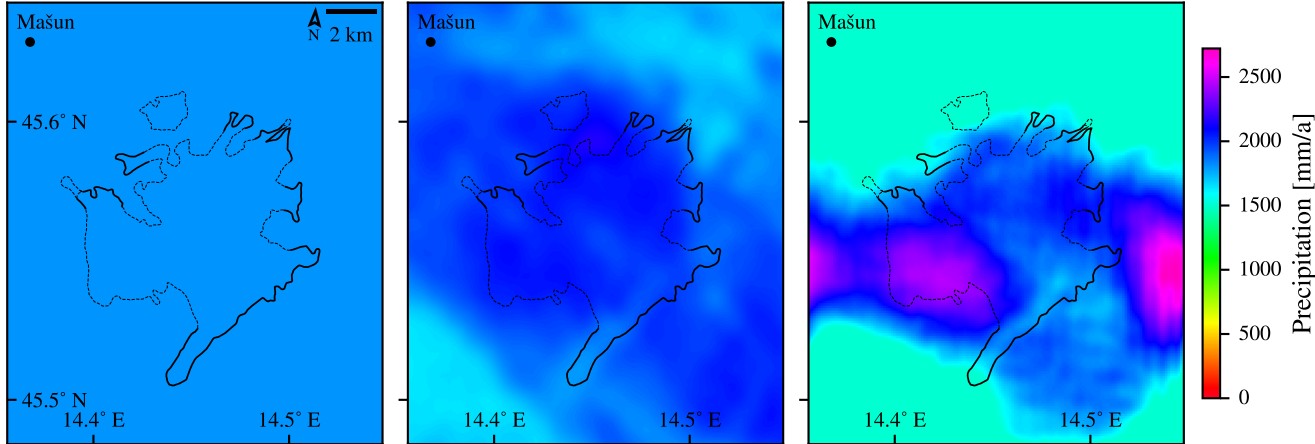

**Figure 5.** Comparison of the precipitation fields generated by the three described precipitation models. The field-based estimates of the glacial limits are plotted as black lines.

the reference average of 1971–1986, to match the simulated glaciation extent with the estimated extent well. This is close to the value of the temperature drop for the larger Alpine area during LGM estimated by Del Gobbo et al. (2023).

### 3.4.2  Precipitation

Three precipitation models were explored in the presented study. The precipitation fields generated by the three presented models are visualized in Figure 5 to demonstrate the great difference they produce for the simulation input.

First is the baseline precipitation model, which assumes uniform precipitations across the domain. The value of the precipitation is taken from the reference weather station and is 1946 ㎜/a. Although this is the simplest of models and not appropriate for large areas, it could work in the presented case, since the simulated area is very limited. Note that the used precipitation value also seems low with regard do the larger Snežnik area with precipitations up to 3000 ㎜/a. The reason for this is that elevation of the station is relatively low, far below the ELA, while the precipitation is expected to increase with elevation. The

precipitation pattern generated by this model is shown on left in Figure 5.

We shall refer to the second model of precipitation as WorldClim model, since it originates in an existing global climate model. We use the WorldClim (Fick and Hijmans, 2017) model of global climate as the source, reduce its coverage to the area of the domain and interpolate the mean monthly precipitation component on a 50 m resolution grid (all of the above was done in software package QGIS). We considered using a local model (Odprti podatki Slovenije) instead, which is based on

data gathered by the Slovenian Environmental Agency (ARSO) in time interval 1981-2010. We found that the local model did not offer any better resolution, while it failed to cover the whole domain, since a considerable part of the presented domain extends across the national border. The two models also notably differ in their prediction over the selected geographical area, which is clearly observable by the mean annual precipitation over the area enclosed by the geomorphologically determined



ice field bounds. For the WorldClim model, the precipitation ranges from 1582 ㎜/a to 1955 ㎜/a with the mean of 1827 ㎜/a in
contrast to the range from 1950 ㎜/a to 2534 ㎜/a with mean of 2294 ㎜/a for the local model. Thus, the local model prescribes
a significantly higher (25 %) precipitation over the critical area for ice field formation. The performed experiments show that
the precipitation from the WorldClim model needs to be increased by a small factor for the resulting ice field to be comparable
in ice area to those of other models (the details are available in Section 4.4). Therefore we assume WorldClim global climate
model underestimates precipitation of the observed geographical area.

Main problem of the climate-model based precipitation model is that its resolution is far from comparable to the target
resolution for the simulations. The best approach to get a better resolution would be to downscale the climate model to micro-
scale, as there are multiple possible methods for doing so (Maraun et al., 2010), however even state-of-the-art solutions can only
reach resolutions of $\approx 1\,\mathrm{km}$ (Karger et al., 2023), which is still far from the required $100\,\mathrm{m}$. In addition, downscaling operation
would require extensive amounts of data, some of which is only available through proxies and simulations, e.g., palaeoclimate.
Therefore, for the purposes of glacial simulation, we currently consider downscaling to micro-scale as infeasible. Instead, we
perform purely mathematical interpolation of the input precipitation field down to the target grid, which is determined by the
target resolution for the simulation at hand. While the interpolated WorldClim model is likely more accurate than the uniform
precipitation model, it does not take enough of the local topography details into account and actually leads to a simulation
result very similar to the uniform precipitation model – see Figure 5 for comparison.

Third precipitation model is based on a physical model, with the measurements only used to estimate some of its parameters.
This is the model of orographic precipitation mixed with a uniform precipitation background. Orographic precipitation occurs
when moist air is lifted as it moves over a mountain range. As the air rises it adiabatically cools, and orographic clouds form
to serve as the precipitation source. Precipitation then mostly falls upwind of the slope that caused the air lifting. Current day
Snežnik experiences heavy orographic precipitation caused by the south–southeastern moist winds that transport air mass from
the nearby Adriatic Sea. Since not all of the precipitation is expected to be of orographic nature, this model mixes it with
uniform precipitation across the domain. The main input to the orographic precipitation model is the DEM of the area, thus the
model output can be of the same resolution as the DEM, without having to invent the high resolution details by interpolation.
On the other hand, the physics of the model are simplified and several model parameters have to be invented instead.

To simulate orographic precipitation, the model integrated in PISM is used, which is an implementation of the model by
Smith and Barstad (2004) and some of its modifications by Smith et al. (2005). Besides the fixed topology, the main driver of
orographic precipitation is wind, which is setup in the model using two parameters – speed and direction. Both parameters are
setup as scalar values – the model therefore calculates precipitation for singular weather conditions, which is likely different
from the average precipitation. Since the past climate is poorly understood we do not attempt to setup these two singular
parameters from observational and palaeoclimatological data but rather derive them experimentally.

Wind direction is the parameter that influences glacier shape the most, therefore, it represents the first step of the study. First,
PISM is setup with insolation adjusted lapse-rate temperature model, uniform precipitation model, and other parameters in a
way that results in a glacier with slightly lesser extend from the geomorphological. Then the simulations are performed with
PISM setup changed to include for the orographic precipitation with a varying wind direction by $30\,°$, constant default wind



speed of 10 ㎧, uniform offset of +3500 mm precipitation (a background precipitation level that can be locally lowered by the orographic precipitation model) and a precipitation factor of 0.5 (which uniformly scales values of individual grid cells of the model output). The listed PISM parameter values can be understood as the background and orographic precipitation shares being at 50%, which is our first approximation.

Simulation results are then analyzed by comparing their final ice cover, as shown in Figure 6. The quantitative results of ice coverage comparison are listed in Table 4 and plotted in Figure 7. Since both sensitivity and specificity should be maximized, the optimal wind angle appears to be 150° (measured clockwise from North), which results in the best trade-off between second highest sensitivity and very high specificity. Such an angle also seems the best from visual comparison of the results (which is admittedly a subjective measure) and is furthermore consistent with the direction of the modern precipitation bearing winds.

| Wind angle | 0 | 30 | 60 | 90 | 120 | 150 | 180 | 210 | 240 | 270 | 300 | 330 |
|---|---|---|---|---|---|---|---|---|---|---|---|---|
| Sensitivity | 0.56 | 0.47 | 0.42 | 0.37 | 0.53 | 0.78 | 0.79 | 0.75 | 0.56 | 0.30 | 0.33 | 0.51 |
| Specificity | 0.56 | 0.47 | 0.59 | 0.80 | 0.65 | 0.62 | 0.54 | 0.52 | 0.58 | 0.70 | 0.63 | 0.58 |
| Sensitivity · Specificity | 0.30 | 0.22 | 0.25 | 0.30 | 0.34 | 0.47 | 0.42 | 0.39 | 0.32 | 0.21 | 0.21 | 0.30 |

**Table 4.** Ice coverage as a function of wind direction.

As the wind direction is selected, other parameters could be optimized further, starting with the most influential ones, the wind speed and the ratio between orographic and background precipitation. We feel that thorough optimization would be counter-productive though, since a large error is already being made by considering a single wind direction and speed. It should be stressed again that the orographic precipitation model does not consider wind angle and speed as the mean values but rather as the exact and permanent values. Consider for example how the uniformly distributed winds between 120° and 180° instead of the single 150° (the average) wind would generate a very different precipitation field. The same would hold even more so for wind speeds, which are in reality likely to be distributed over a much wider range of values than the wind angle. Therefore we keep the default wind speed of 10 ㎧ and other parameters of the model (details in Table 2)

Selection of the relative magnitudes of background and orographic precipitation terms was done as follows. Background precipitation was set to equal the measurement at weather station Mašun (1946 ㎜/annum), since the latter lies in an area where orographic precipitation does not generate any additional precipitation and therefore depends on background precipitation only (See Figure 5). Then the orographic precipitation was added unmodified from the model, resulting in average increase of 323 ㎜/annum and a maximum increase of 1526 ㎜/annum across the domain. The resulting precipitation is then 2270 ㎜/annum on average with a maximum of 3475 ㎜/annum, which is in line with the estimated modern precipitation of up to 3000 ㎜/annum on average for the Snežnik plateau. Therefore we treat the precipitation field generated in the way described above as satisfactory and do not optimize it further.

To summarize, the orographic precipitation model assumes a prevailing wind originating from 150°, wind speed of 10 ㎧, 1946 ㎜/annum of background precipitations and orographic precipitation model with default parameters as provided by PISM.



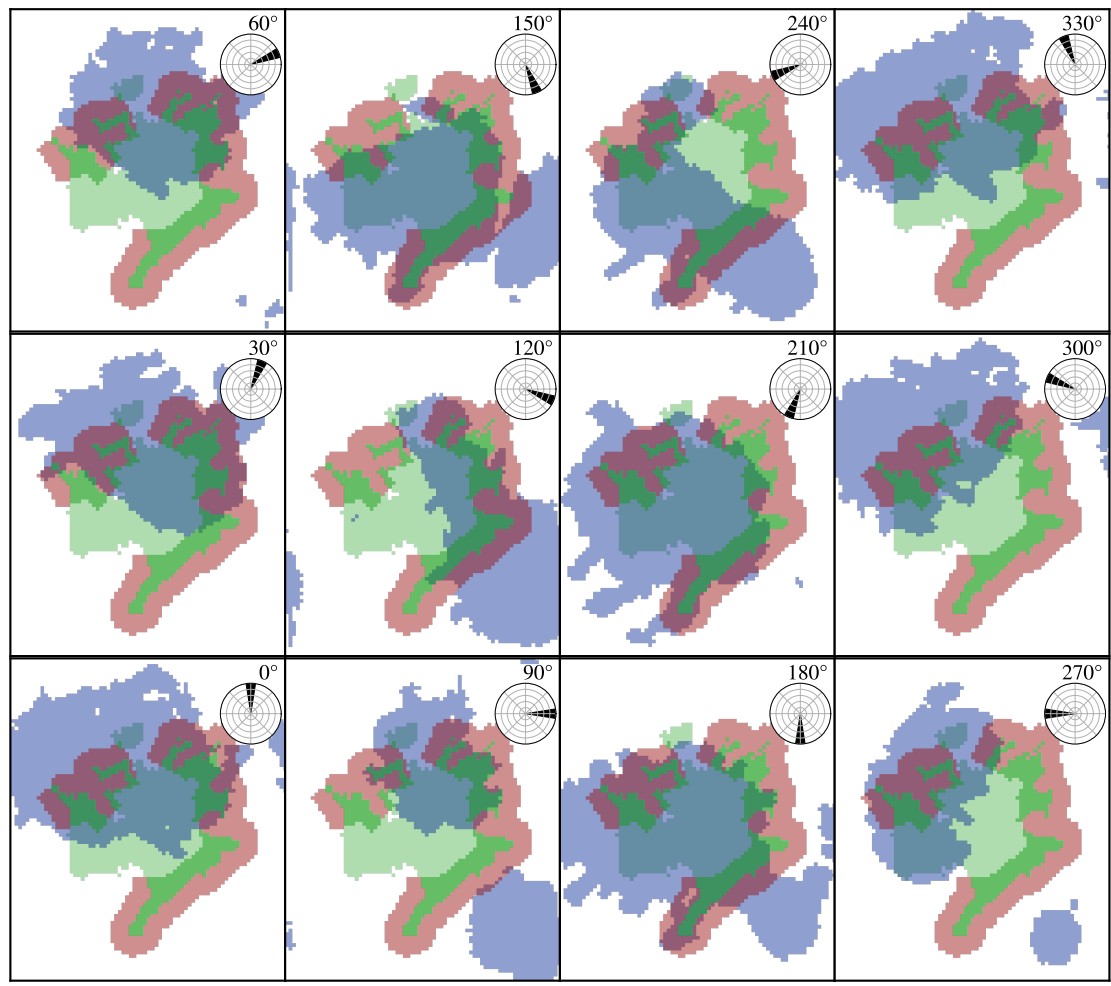

**Figure 6.** Influence of the prevailing wind direction on glacier formation. Wind direction is marked in top right corners along with wind roses. The resulting ice coverage pattern is calculated hte presented orographic precipitation model in combination with the adjusted lapse-rate temperature model.



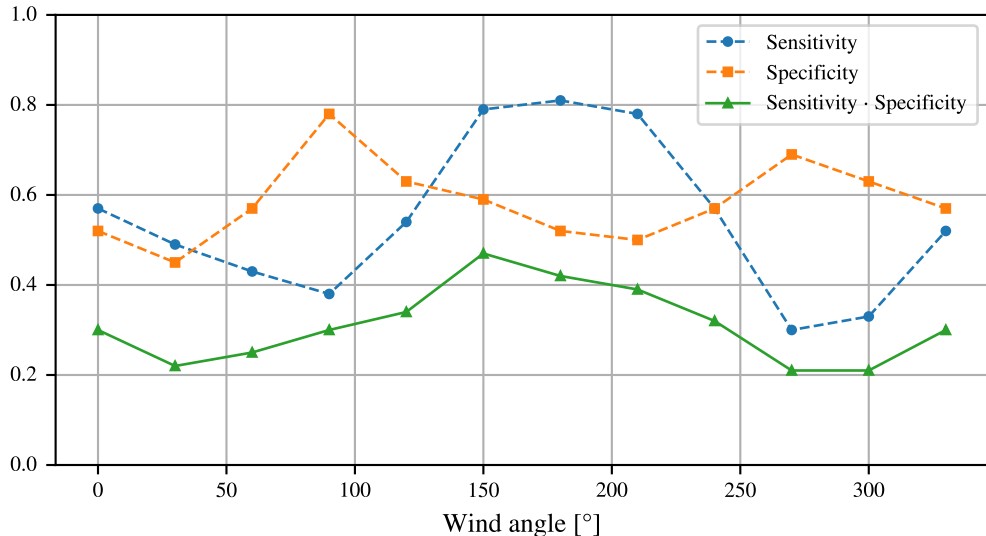

**Figure 7.** Quantitative metrics for the simulation results of the wind direction parameter sweep. The highest value of Sensitivity multiplied by specificity is for wind angle of 150 °.

In a final note regarding the orographic precipitation model, we should acknowledge that the model was created with different setting in mind. Namely, in the direction from which the wind blows should preferably be a body of water or at least have uniform elevation and as such also uniform starting moisture content. These two assumptions are gross oversimplification in our case. The domain could be extended in the direction of the wind to dilute the effect of this error because of the increased distance between the area of interest and the wind origin, but that would increase the error caused by other model assumptions and simplifications. We have performed a preliminary study of how the domain size influences the precipitation pattern over the area of interest. Although the details of the preliminary study are out of scope, its results show that changing the domain extent significantly alters the precipitation pattern but the pattern does not converge when increasing the domain size. Therefore, we use the orographic precipitation model on the presented domain as a demonstration that orographic precipitation is a likely candidate for the observed glacier shape but we cannot claim so with certainty nor can we claim that we discovered the exact palaeoclimatic parameters.

From Figure 5, the orographic precipitation model can be seen to stand out with highest variability across the domain. The patterns of other two models are quite similar within the geomorphological ice field area, which causes the simulation results to be also quite similar, as will be shown later in the paper.

### 3.4.3 Annual cycles

The above described climate models are all used to model annual mean temperature and precipitation. The exception may be WorldClim precipitation model, which can be built to model precipitation at any scale supported by the Worldclim dataset.



PISM is adaptible in the way climate data is supplied, and can easily work with annual mean data, however, we found in
preliminary experiments, that simulation results differed greatly when more detailed data, e.g. monthly means, was used instead
of the annual data. The likely reason is monthly variability in precipitation, which peaks in early winter, as shown in Figure 3.
Consequently, we opted to use monthly means for climate data inputs and to that end we apply annual cycle model on top of
previously described models.

Annual cycle model averages monthly values of temperature and precipitation from the reference weather station data and
applies them on the temperatures and precipitations modelled by previously described models. The exception being WorldClim
precipitation model, which we constructed of 12 monthly precipitation fields instead of the annual field for the domain and
therefore provides monthly values implicitly.

The following procedure is applied to build the annual cycle model. First, monthly mean of temperature and precipitation are
taken from the raw weather station data (Figure 3). Then the monthly mean temperatures are converted into monthly offset from
annual mean, and monthly precipitations into monthly factor of annual mean. Finally, temperature offsets are applied on the
selected temperature model output using PISM's "temperature offsets" and precipitation factors on the selected precipitation
model output using PISM's "precipitation scaling".

## 4 Results

In this section we present the results in several forms. First we confirm that the ice field volume converges and we provide a
time estimate for the convergence to complete. Then we explore how different temperature and precipitation models influence
the simulation results and we comment on which models seem best to use on the presented area. Finally we present the climate
under which we find the optimal conditions for the growth of an ice field in the form that was geomorphologically established.

### 4.1 Climate setup

The primary objective of this study is to identify the climatic parameters that would allow an ice field atop the Snežnik
plateau to align with the extents determined by geomorphological evidence. While this goal will be reached after a large set
of experiments have been performed and analysed, a similar goal is set for the first step of experimentation. To even start
experimenting with different computational models, since glaciers are not present in the area today, favorable conditions for
formation of glaciers on the domain need to be established first. I.e. the first goal is to setup climatic conditions that would
allow for formation of an ice field with a similar to the geomorphologically estimated extent.

Using the simplest two models – the lapse-rate temperature model and the uniform precipitation model – a set of simulations
has been performed to find suitable starting conditions for further exploration. In Figure 8 the results of a small grid search
with air temperature spacing by 0.5 °C and precipitation spacing of 10 % are shown. From the results, several conditions that
result in conforming ice extent can be chosen; we use the offset of -6 °C compared to modern temperatures and absolute annual
precipitation of 2041 mm/a as a starting point for further experiments.



**Figure 8.** Influence of the mean air surface temperature and mean precipitation on the formation of the glacier on the presented domain. Both parameters can be used to either decrease or increase glacier extent. Furthermore, the results on the diagonal are very similar indicating that the parameters are not independent, increase in temperature is equivalent do decrease in precipitation, at least on the range of values from the study.





More importantly though, an unexpected finding can be made from the results shown in the figure. An decrease in temperature of 0.5 K coupled with a 10 % decrease in precipitation does not significantly alter the extent of ice field. While these are just human friendly numbers that were used to setup the parameter scan experiments, we can nevertheless write our observation in a general form:

$$
\begin{aligned}
I(T, P) &\approx I(T - k_T, k_P \cdot P) \\
k_T &\approx 0.5 \\
k_P &\approx 0.9,
\end{aligned}
\tag{2}
$$

where $I$ is the ice field that depends on surface air temperature and precipitation fields $T$ and $P$, respectively, and $k_T$ and $K_P$ are the coefficients representing approximately a 0.5 K increase in temperature and a 10 % decrease in precipitation. We shall refine the parameters in further experiments, after the confirmation of simulation convergence and selection of best climate models.

## 4.2 Convergence

Although various models for temperature and precipitation were used in the preliminary experiments, we noticed only insignificant influence of the model selection to the convergence of ice volume and ice extent. Therefore the same settings were used for all the simulations, including the selection of simulated time limits, and the number and resolution of simulations within grid sequencing. Figure 9 shows the progression of ice volume from several experiments that were used to determine the required simulation time and the optimal resolution for further simulations. Note that the shown lengths of simulation on each individual grid size are different from the values specified in Section 3.2, which were selected based on the simulation shown here and other similar experiments, to optimize for low execution time and satisfactory resolution of the resulting ice cover. From this figure, the convergence of solutions towards a steady-state glaciation can be seen, depicting both the time needed for ice accumulation (about 2500 years) and the final volume after it converges (about $2.6 \cdot 10^{10}$ m³). It can be noted that the resolution of 400 m is not detailed enough while other tested resolutions all converge to the same volume of ice. A switch from 400 m to 200 m could also be done sooner, at 500 to 1000 years to allow for faster overall convergence. Also, the variation of ice volume after it has converged can be clearly seen. This means that all further analysis of ice fields, that have been performed on a snapshot of the simulation in its final time step will not be perfect, since it is likely that those will focus on slightly different points within the range of natural variation.

## 4.3 Temperature models

In this subsection, the previously defined temperature models (the lapse rate and the insolation adjusted lapse-rate model; see Subsection 3.4.1), are visually and quantitatively compared, based on simulation results. Experimental simulations are setup to mimic modern precipitation levels and -6 °C offset from modern temperatures. In Figure 10, the insolation adjusted lapse-rate model is used with three values of parameter $A_S$. Value $A_S = 0$ makes the model behave identically to the lapse-rate model, since the insolation adjustment is multiplied by zero. Then the value $A_S = 1$ represents the case whee solar insolation





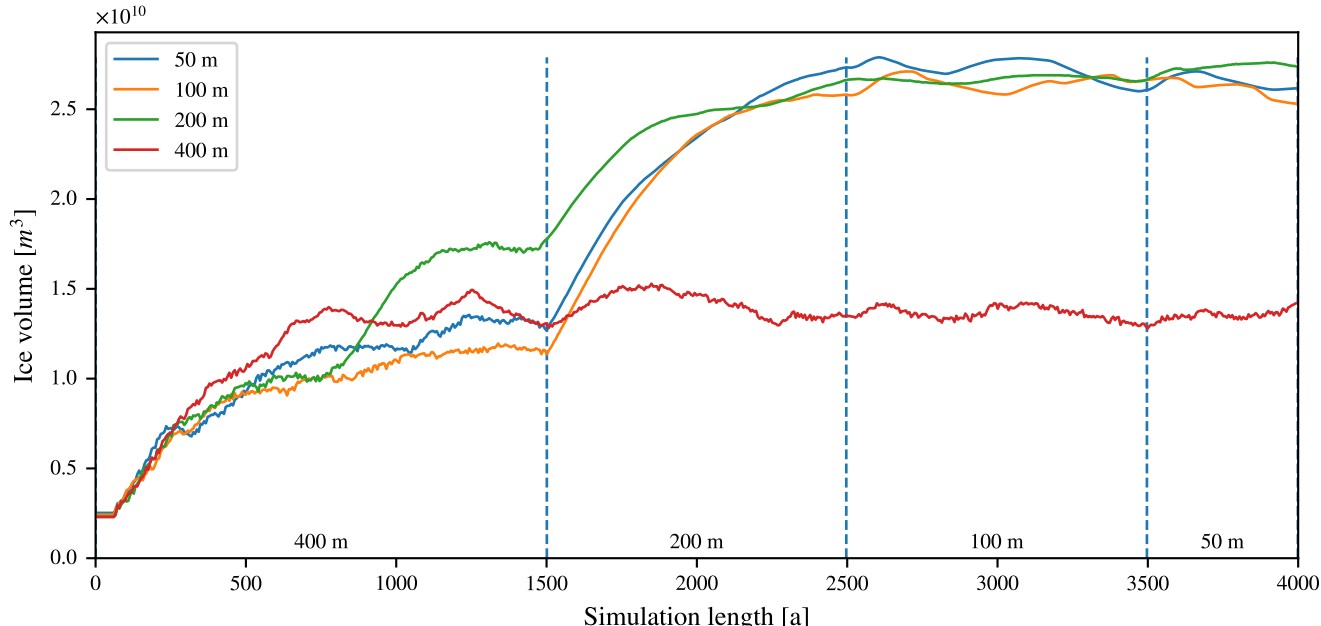

**Figure 9.** Convergence of the ice field characterized by the ice volume. Presented are four experiments, all using grid sequence approach for faster execution, starting with resolution of 400 m and sequencing up to resolutions of 400 m, 200 m, 100 m, or 50 m (the label in the legend). Grid sequence resolutions are marked on $x$ axis, along with their transition times (blue vertical dashed lines), since all experiments use these to optionally switch resolutions. However, each experiment only transitions up to its final resolution and then continues execution until the 4000 a limit of the experiment.

amplitude is set to mimic the observed ELA difference of 150 m between insolated and shaded slopes. Finally the value $A_S = 3$ represents an overvalued effect of insolation.

We experimented more with the insolation adjustment than is shown in Figure 10 but found no significant effect until the
$A_S$ is raised very high, e.g. about 5 times its expected value. Similarly, the effect is entirely hidden in the noise if the objective function is observed instead of observing the results visually. Therefore, while our analysis indicates that selecting either of models would suffice for our use-case as they cause only minimal differences in simulation results, we selected the more realistic one, that is the insolation adjusted lapse-rate model with its amplitude parameter $A_S$ set to its most likely value of 1 for further simulations.

**4.4   Precipitation models**

In this subsection, the differences between the implemented precipitation models (see Subsection 3.4.2) are explored. Selection of the right precipitation model proved to be the most significant one regarding the overall shape of the simulated glacier.



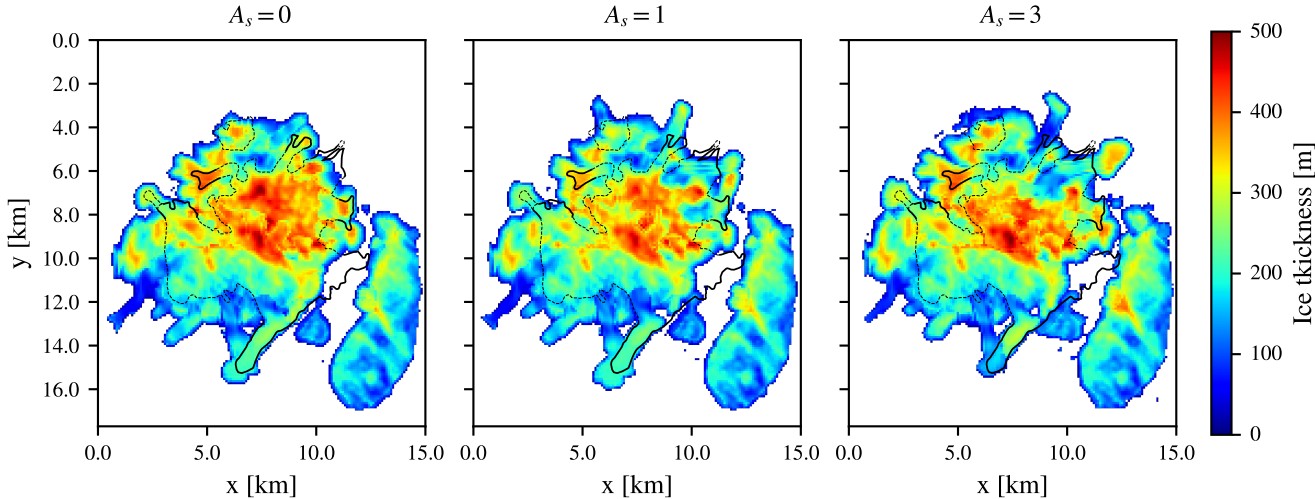

**Figure 10.** Comparison of the simulation results that use the two presented temperature models. The geomorphological area of the glacier is outlined in black. The results were obtained by using the insolation adjusted lapse-rate model with three values of $A_S$ as depicted above the plots. The model with $A_S = 0$ equals pure lapse-rate model, since the insulation adjustment in that case equals 0 everywhere on the domain.

Comparison of the simulation results obtained by using uniform, WorldClim, and orographic precipitation models is shown in Figure 11. All three models are setup to mimic modern precipitation levels and -6 °C offset from modern temperatures.

While the quantitative analysis does not show large difference, both constant and WorldClim models cause the ice to grow askew towards the north compared to the geomorphological extent. Under these two precipitation models the variation in precipitation is very low and the existence of ice is determined primarily by the temperature model, which in turn is driven by domain elevation. The domain elevation is skewed towards the north compared to the geomorphological ice field position and thus so are the simulation results. It should be stressed that the observed north-south imbalance is in addition to and much

larger than the one caused by the insolation adjusted lapse-rate temperature model. Only the orographic precipitation model transfers the distribution of ice southward via precipitation redistribution.

    Unlike the visual inspection, the value of the objective is only slightly higher for the orographic model (0.32) than for the other two models (0.26 and 0.31). The larger ice field located to the southeast is partially responsible, which covers the largest part of forbidden area in the simulation with orographic precipitation model. The ice sheet to southeast is otherwise known but

was excluded from the performed study, since the isolation of the two ice fields was clear from the performed geomorphology studies. This case therefore highlights a disadvantage in the presented quantitative measure - the unmarked neighbouring ice fields may have a significant impact on the accuracy of the qualitative measure.

    Nevertheless, orographic model is best both visually and quantitatively and is selected as the base precipitation model to be used for further simulations of this geographical area.



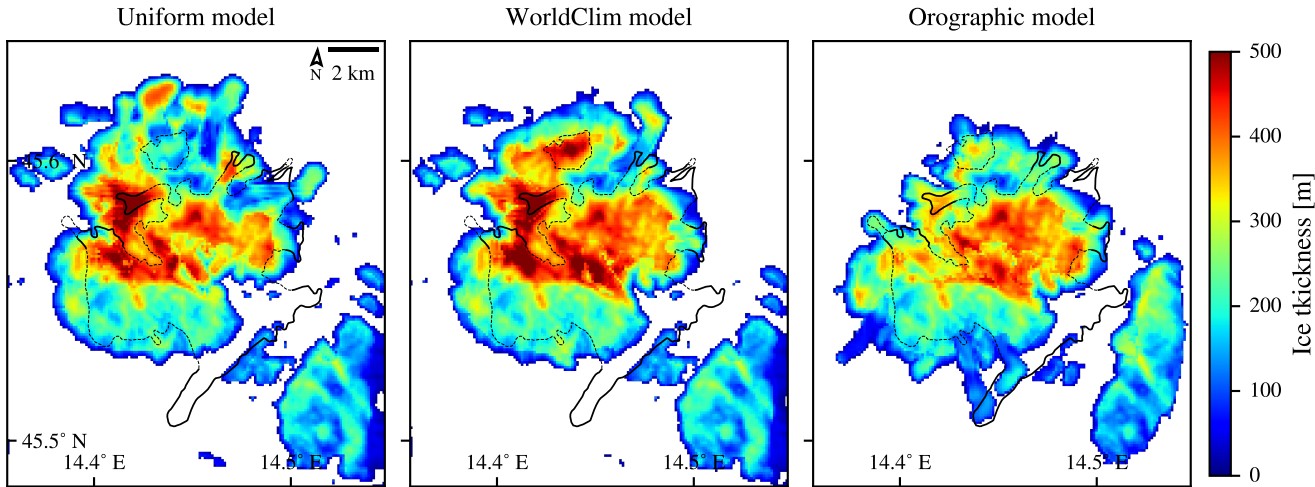

**Figure 11.** Comparison of the results of simulations that use the three presented precipitation models. The geomorphological area of the glacier is outlined in black. Values of the objective for uniform, WorldClim, and orographic models are 0.26, 0.31 and 0.32, respectively.

## 4.5 Climate conditions for Snežnik

In this subsection we present the results that drectly address the goal of the study – improving our understanding of the past climate-glacier dynamics at the Alps-Dinarides junction. The results are in the form of a linear relation that describes the best conditions for the formation of ice field on plateau Snežnik and the ice field simulated with one set of ideal conditions on the domain with resolution of 50 m.

First, an experiment has been prepared to obtain the best climate conditions, i.e., the conditions under which the simulated extend matches the geomorphological extent the best. Since the temperature and precipitation have been found to be related, a fine-grain grid search has been performed on air temperature only, with the precipitation fixed to modern values. The main results are summarised in Figure 12 in form of the objective values for all performed simulations and in Figure 13 in form of ice extents of the nine best performing simulations. Visually, the shown extents are difficult to order by quality. Some of the ice field features, e.g. termini, are better covered at higher temperature while others at lower temperature. Looking at the value of the objective, small differences can be made out and the objective value peaks at -5.6 °C surface air temperature offset, although the differences compared to other offsets are small. The result would be perfectly acceptable if the air temperature is either increased or decreased up to 0.2 °C, only at different trade off between sensitivity and specificity.

Using the obtained set of best climate conditions, the simulated ice field extent is examined in more detail. The resolution of 50 m is computationally too expensive to be used for exploration of the climatic conditions while providing results that are only marginally better then those obtained by 100 m resolution. It has been selected for the final simulation along with the climate conditions defined above, since it provides some additional detail, especially in the ice flux. Ice flux in the final state of the simulation is shown in Figure 14. The simulation shows that the ice flow patterns on Snežnik are largely controlled



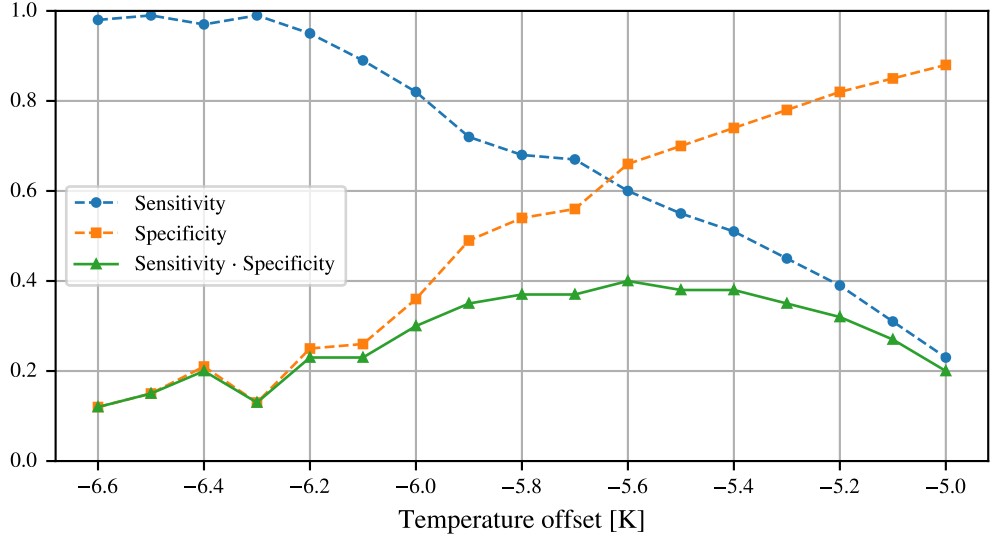

**Figure 12.** Value of the objective for the fine-grain sweep over the temperature with a fixed precipitation value. The highest value of the objective (sensitivity multiplied by specificity) is at -5.6 K.

by subglacial topography. The glaciated area is composed of two plateaux above 1300 m that behaved as accumulation areas, from where the ice was flowing almost radially downstream. The ice from northern and southern plateau joined in a large karst depression in between them, from where it was drained by an outlet glacier flowing towards south/south-east. This is in agreement with geomorphological evidence despite of some discrepancies between simulated and geomorphological ice extent.

Lastly, we look into the relationship between temperature and precipitation. As seen in Figure 8, and described in Eq. 2, any increase in precipitation can be countered by a decrease in temperature to keep the conditions for simulated glacier formation about the same. Using the best temperature and precipitation models, we have performed additional experiments to improve the approximations for coefficients of Eq. 2, and found that the values of $k_T = 0.45$ and $k_P = 0.9$ are nearly optimal. Based on the Eq. 2 and the experimentally discovered optimal coefficient values, the final equation describing the optimal conditions can be written with $x$ being the free parameter:

$$
\begin{aligned}
T &= -5.6\,\mathrm{K} - x \cdot 0.45\,\mathrm{K} \\
P &= 1946\,\mathrm{mm/annum} \cdot 0.9^x \\
x &\in \mathbb{R} \text{ and } x \text{ is small}
\end{aligned}
\tag{3}
$$

To confirm that climate conditions that conform to Eq. 3 result in similar ice fields, we have designed an additional experiment. For a limited set of values $x$, the ice field is simulated and the results are compared to the results simulated at $x = 0$





**Figure 13.** A fine-grain sweep over a single parameter – near-surface air temperature. The objective measure for the ice cover is given in Figure 12.



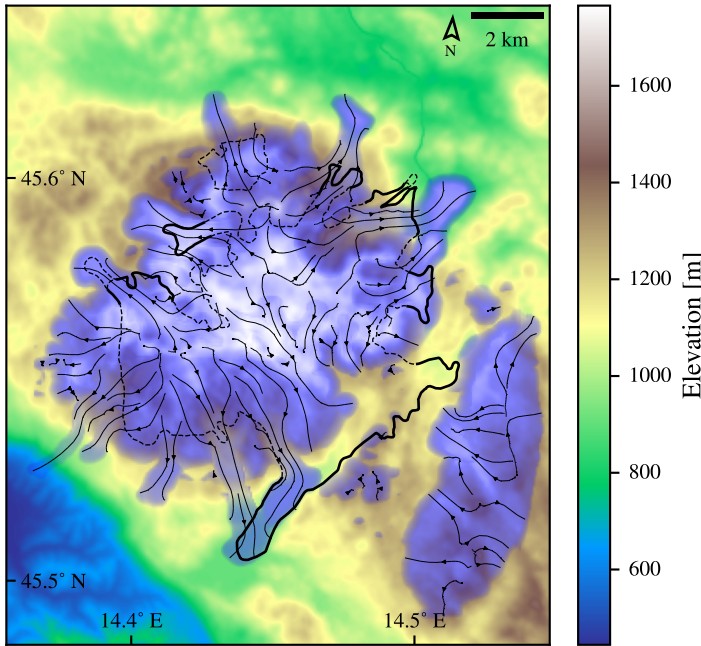

**Figure 14.** Streamline plot of the ice flux (arrows), overlaying the terrain map and the geomorphological ice field outline (thick lines for the clear and dashed lines for the unclear outline). The experiment was performed on domain with 50 m resolution.

according to the defined objective. The results of the experiment are presented in Figures 15 and 16. Besides confirming the equation, the described experiment also gives a range of valid values for $x$, where Eq. 3 holds.

The range of $x$ seems to be on the interval $[-3, +8]$ from the selected reference point. Therefore one can take a valid value of $x = 3.11$, which is within this interval to get the temperature offset of -7 °C, and precipitation factor of 0.66 relative to modern conditions. Adjusting the temperature offset by previously mentioned 0.4 °C of difference between our modern reference interval and the pre-industrial era, the temperature offset exactly matches the results of Del Gobbo et al. (2023), who reconstructed the LGM temperature to be offset by -6.6 °C relative to pre-industrial era. The precipitation of 0.66 relative to modern reference interval is also close to the reconstructions by Del Gobbo et al. (2023), where the precipitation for the general area is estimated to lie between -10 % and -30 % relative to pre-industrial.

## 5 Conclusions

A qualitative metrics of the simulation quality was developed, which can take clear and unclear geomorphologically deduced ice boundaries into account. Several simulations resulting in an ice field with climatic forcing compatible with LGM conditions were successfully performed. The factor with the largest effect on the simulation results was found to be the precipitation pattern. Global precipitation models, such as the one provided by WorldClim were found to be insufficient for simulation of



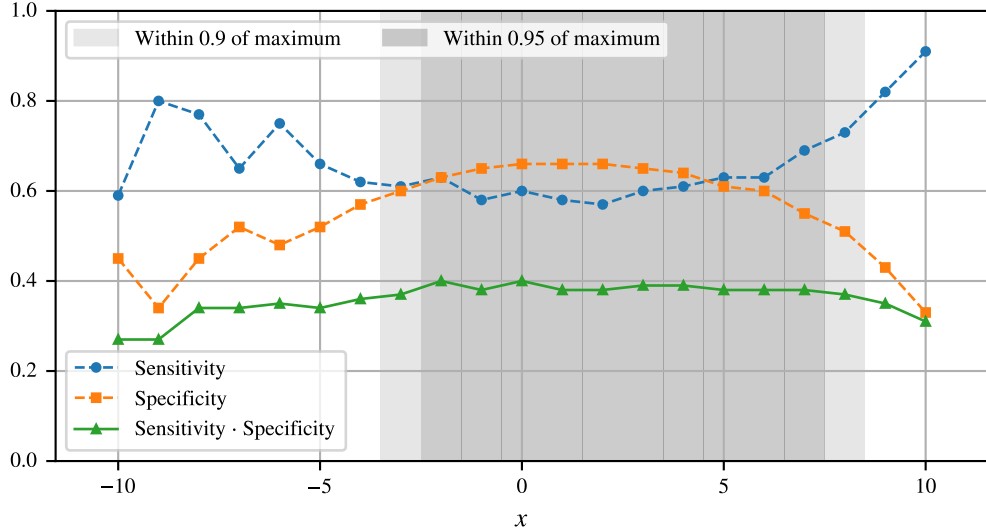

**Figure 15.** Relation between the free parameter of Eq. 3 and the objective value. The shaded areas represent solutions where the value of objective function is within 90 % and 95 % of the maximum obtained value.

accurately shaped ice fields. On the other hand, even the simplest temperature models, such as one based entirely on elevation
and lapse rate were sufficient for our use case.

This research has successfully established a quantitative framework for the assessment of palaeoglacial simulations that integrates both definitive and provisional geomorphological deduced ice boundaries to improve the accuracy of model results. The qualitative metrics developed here help to determine the agreement between model-derived ice extents and those derived from geomorphological field data. The series of simulations carried out under the climatic conditions of the Last Glacial
Maximum have produced an ice field that is consistent with empirical geomorphological reconstructions.

An important finding of this research is the disproportionate influence of precipitation patterns on the simulation results. It was found that the spatial distribution of precipitation, especially when influenced by orographic factors, is crucial for the accurate representation of glacier extent. It has been shown that global precipitation models such as WorldClim do not have the necessary resolution nor the orographic sensitivity to accurately simulate the shape of ice fields. This inadequacy requires
the integration of high-resolution topographic data and locally refined precipitation data to capture the complex interactions between topography and climate. We must bear in mind that the orographic precipitation model is based on assumptions about wind patterns that may not accurately reflect the complex interactions between topography and climate, especially given the lack of palaeoclimatic data on wind direction.

Conversely, the adequacy of elementary temperature models relying solely on lapse rates and elevation data reveals the
relative insensitivity of ice field extent to the temperature data in this study. This observation points to a potential area of computational optimisation in modelling efforts where high-resolution temperature data are not readily available.



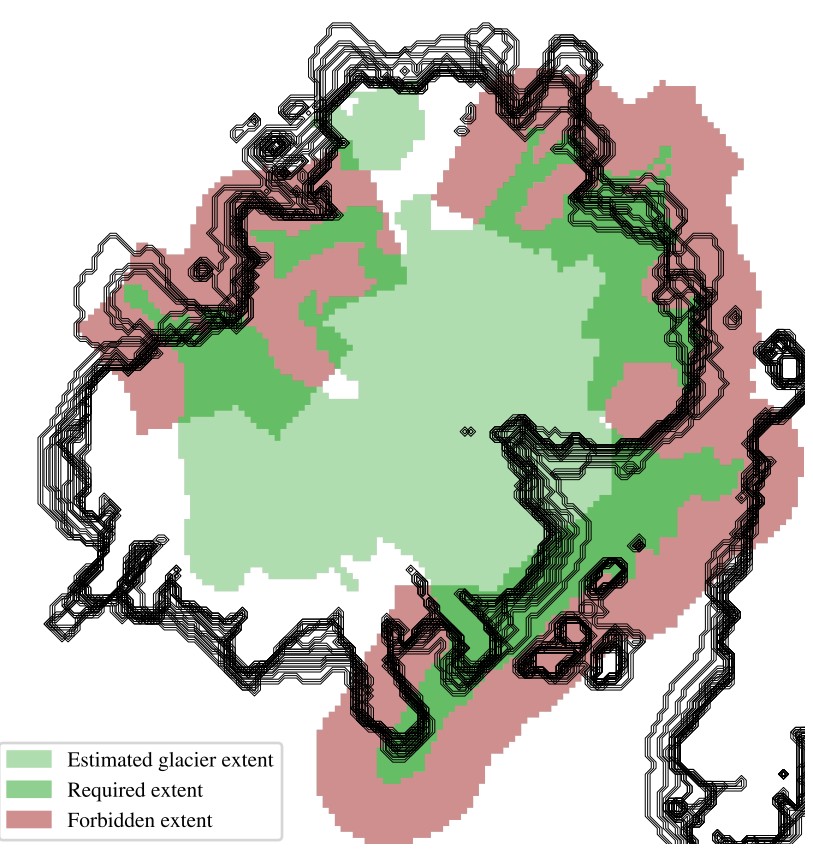

**Figure 16.** Area of simulated ice fields for the values of $x \in [-3, +8]$, for which the value of objective function is within 90 % of the maximum obtained value.

The study reveals that air temperature and precipitation are closely linked when it comes to the size of glaciers. These two factors affect each other in a way that creates a balance within the simulations. This means that different combinations of temperature and precipitation can result in similar glacier extent. This interplay precludes the determination of exact climatic conditions based on glacier morphology alone and suggests a broader framework of possible past climates that are consistent with the observed glacier extents. The main achievement of this research is the derivation of a straightforward model that connects surface temperature and precipitation data and provides a simple but robust approach to simulating the ice fields of the LGM. This model is consistent with geomorphological field data and is a convincing demonstration of the effectiveness of integrating simple climate models with detailed precipitation data.




Although advanced, the presented climate models used may still be an oversimplification of actual past conditions. The temperature model could be improved by using mountain shading in addition to insolation adjustment and orographic precipitation model could be extended to include a set of most typical wind conditions. There is also the option of extending the till and hydrology models based on additional field observations to better simulate the karst conditions of the researched geographical area.

*Code and data availability.*  Code and data will be made available on request.

*Author contributions.*  GK conceptualized and supervised the work. MŽ and US performed geomorphological investigation, resources and validation. MD provided the methodology, software and performed visualization and software investigation. The whole team participated in manuscript writing

*Competing interests.*  The authors declare that they have no conflict of interest.

*Acknowledgements.*  The authors would like to acknowledge the financial support of the Slovenian Research And Innovation Agency (ARIS) research core fundings No. P1-0419, P2-0095, and P6-0229 and project funding No. J1-2479.



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
