# Peer review of "Simulation of a former ice field with PISM – Snežnik study case"

_EGUsphere, 2024_

## Referee Comment (RC3)

[referee-annotated manuscript omitted]

---

## Author Response (AR1)

**Reviewer 1:**

We will answer the remarks by breaking them down and answering them one by one. To improve readability, we shall format the original remarks in red, and excerpt from the modified manuscript in blue. We shall also attach a pdf of the manuscript with changes clearly marked.

This manuscript presents simulations of a small plateau glacier in present-day Slovenia during the last glacial maximum (LGM). This work is motivated by earlier geomorphological reconstructions of past glaciation. It aims to answer the question whether these reconstructions can be used to estimate the regional climatic conditions during the LGM.

While the study mostly fails to provide a robust reconstruction of glacial climate, it succeeds in establishing a methodology that may be used for similar attempts in the future. Despite the large uncertainties inherent to paleoclimate studies, the manuscript describes a solid methodology and and a mostly complete discussion of its shortcomings. The individual steps of the modeling approach and the evaluation are described in great detail and the text follows a clear logical structure. The figures support the text optimally. I greatly enjoyed reading this well-written piece of work.

I have three major remarks:

1) I think the main contribution of this study is not the actual temperature reconstruction, because it is rather imprecise and lacks an uncertainty estimate. Instead, I concur with the authors that "This research has successfully established a [..] framework for the assessment of palaeoglacial simulations that integrates [..] geomorphological deduced ice boundaries to improve the accuracy of model results" (l471ff). I think the discussion of results and the conclusions should reflect this change of emphasis and provide a deeper discussion of the shortcomings as well as how they can be ammended.

1) Considering both reviewer remarks, we will attempt to word this better and to rebalance the focus on different achievements in the manuscript. First, to explain our position, we value the presented climate construction above the proposed framework. The reasoning is that the climate reconstruction confirms the latest LGM reconstructions without being specifically tuned on them (apart from the free parameter that tunes the precipitation/temperature balance). I.e., our study is an independent validation. The framework on the other hand was very helpful in our study but does not seem to be very general. Although we hope somebody will find it inspirational and derive their own tool based on it, we view it mostly as a tool that was born out of necessity and is highly focused on our particular case.

We certainly did not stress the framework shortcomings enough. We shall add this paragraph to the revised manuscript:

The framework alleviated the difficult task of sorting simulation results by quality but did not eliminate visual checks entirely. The reason lies in its shortcomings, which we list here. First, its parameters require setup, which in turn requires experimenting. In the presented case the experimenting was light, but could have proven more difficult for a more demanding ice field shape. Second, the framework is missing a methodology for ignoring neighbouring glaciated areas. We expect neighbouring glaciations can often be a problem since the simulator requires the domain to be rectangular and to be somewhat larger than the area of interest. Thus, it is

likely for most studies to find their domains contain some glaciers that are outside the focus and require special treatment in analysis. Finally, the proposed framework is very specific in its demands for at most two types of limits -- clear and unclear. There is currently no room for quantifying the clarity nor for including more limit types. While this framework would work if only clear limits were given, such cases could also make use of simpler analysis methods.

A few open questions include the choice of surface mass balance model (PDD) and in particular how the PDD factors were chosen. Are they left at their default values or were they adjusted to this particular domain?

We shall add the PDD factor values to the tables:

- factor_ice (0.00879121 meter / (Kelvin day)) = (8 mm liquid-water-equivalent) / (pos degree day)
- factor_snow (0.0032967 meter / (Kelvin day)) = (3 mm liquid-water-equivalent) / (pos degree day)
- refreeze (0.6)

These values equal PISM defaults and simulations were found to be very sensitive to them in previously published studies. We did perform some preliminary (i.e. not on the final set of models and parameters) sensitivity studies on several other parameters and obtained similar results as other studies did. Our baseline for sensitivity analysis was significantly different from the shown results (e.g. different precipitation/temperature ratio, different climate models and their parameters), thus we avoided adding specifics on the sensitivity analysis. However a near complete omission of discussing sensitivity is an oversight on or part and we shall add a paragraph to the manuscript, section "Conclusions" (which shall be renamed to "Discussion and conclusions"):

Another area where improvements should be sought is in better determining various unmentioned model parameters. Within the preliminary analysis of climate models we explored the sensitivity of the simulated ice field area and volume with varying modelling parameters such as those related to the domain grid, ice rheology, stress balance, basal sliding, and till properties. Our findings are consistent with those from other studies published by Žebre et al. (2021) and Candaş et al. (2020). Specifically, the simulated glaciation extent and volume are as sensitive to choices related to parametrization of other models as they are to climate models. This sensitivity suggests that small variations in parameters can lead to significant differences in the results. Given this, along with the lack of local measurements to aid in parameter adjustments, the presented results should be interpreted with due caution. For future work, a methodology to set up or optimise all the major model parameters should be developed.

How exactly is the temperature reconstruction impacted by the incomplete knowledge of precipitation?

Temperature and precipitation models are treated separately. Neither is impacted by the other in any way. While not very realistic, such an approach is rational given all the unknowns and the known temperature/precipitation relation. We shall add to the manuscript:

Within this study, temperature and precipitation models are treated separately, with one having no influence on the other.

Why does the model consistently fail to simulate certain parts of the reconstructed glacier area?

Since not all the performed simulations are shown in results, the article might be a bit misleading. In reality, all parts of the ice field can be simulated, given enough tweaking of the climate models. However, we were never able to simulate them all within the same simulation run. The presented results show only some of the best trade-offs between overrepresentation on one, and underrepresentation on another section of the prescribed ice field limits. For example, the eastern-most part of the glacier is not in any of the shown results but can become even over-expressed relative to other parts of the ice field if different wind direction is chosen to govern the orographic precipitation.

2) Virtually all simulations include a substantial glaciation of the mountain to the southeast of Snežnik, which I believe is called Ceclje. This is not too surprising as the elevation of the surrounding terrain is similar and so are the climatic conditions. However, the geomorphological reconstructions show a strict boundary and thus a "forbidden area" that penalizes the simulations for having ice there. How certain are you that Ceclje was not glaciated during the LGM? Should the possibility of its glaciation not be reflected in the skill metric? And, more generally, the manuscript takes the approach of using the geomorphological data to inform the modeling, but the opposite could also be done. I think the possibility of a glaciation to the southeast should at least be discussed.

The SE part, so-called Ceclje, was probably glaciated as well, but was part of a different ice-field, called Gorski Kotar that has not been included in our analysis. This has been presented in some previous works of some of the co-authors of this study (e.g., http://dx.doi.org/10.1016/j.geomorph.2016.01.005, http://dx.doi.org/10.1080/17445647.2015.1095133). According to these previous studies Snežnik and Gorski Kotar ice fields were not connected, which was found out via provenance analysis of proglacial deposits in the Gomance area. The forbidden area along that stretch therefore serves in the analysis of Snežnik glacier alone, however the presence of Gorski Kotar glacier certainly creates errors in the automated analysis. Our previously listed addition to section "Discussion and conclusions" addressed this problem.

3) Lastly, I would like to see a discussion of how the reconstructions could be improved. Ideally, geomorphologists and other field-going scientists could used the simulations results to inform their work and thereby help to better constrain the next iteration of the reconstruction. What kind of data and from where would be most valuable?

Data from karst depressions filled with glacigenic sediments would be potentially useful to simulate karst conditions and therefore aim to better understand the interaction between glacial and karst processes. Karst depressions that were at the edge of glaciation (e.g., Gomance, Praprotna draga, Grda draga, Črna draga) would be the best candidates for such studies.

We will add the following to conclusions:

On the other hand, the existing simulations can be used to find additional evidence of glaciation in the field to better constrain the next iteration of the simulation. While a larger area of Snežnik has already been mapped and every standard-sized landform has been mapped, there is still much that is unknown when it comes to the sediment filling of karst depressions. The existing best-fit simulations could potentially help locate locations for further research, e.g. to plan drilling

into the sediments trapped in karst depressions close to the simulated ice boundary.

Minor remarks:

figures: I would prefer the original model grid to be visible in all figures and therefore recommend not using interpolation in figures 4 and 5. I like the pixelated nature of figure 6 etc. better.

There is no interpolation used in any of the figures. There are, however, different resolutions used. These are caused by different data being natively (within our study) in different resolutions. For figures depicting simulation results, the resolution is limited to the resolution of results, and is either 100 m or 50 m. Topography on the other hand is shown in its native resolution, which is 25 m. Temperature models are developed from the topography and follow its resolution. WorldClim precipitation model is (externally) interpolated (from a resolution much lower than 100 m) to the resolution matching the topography (25 m) in which it is then shown. Orographic model on the other hand is calculated within PISM and its resolution matches the resolution of results, which is 100 m.

The choice of using native resolutions everywhere seems the most natural since the inputs are displayed in the form that can be used by PISM (which can process inputs of higher resolution than it is simulating on), outputs are also displayed in the resolution used by PISM, and data is not modified for the purpose of plotting figures.

We will include some explanation regarding the resolutions used in the figures themselves.

line 3: There is no need to justify the use of PISM. I would remove ", which is an established...".

Agreed, we shall simplify the statement as proposed.

line 9: "Snežnik" is not yet introduced and it is unclear what it means at this point.

Agreed, we have introduced Snežnik in the abstract:

In this paper we present a reconstruction of climate conditions during the Last Glacial Maximum on a karst plateau Snežnik, which lies in Dinaric Mountains (southern Slovenia) and bears evidence of glaciation.

line 46: I think the introduction should include one paragraph with a preview of the main results.

We will conclude the introduction with the following two paragraphs.

Through the use of orographic precipitation coupled with a simple elevation-based temperature model, we manage to simulate ice field distributions that conform better to the geomorphological evidence. We find optimal overall precipitation and temperature offsets relative to modern values to be a close match with the established estimates of local precipitation and temperature in the LGM, thus giving more evidence to these estimates. We are, however, unable to credibly simulate the finer details of the ice field, such as smaller outlet glaciers.

As a part of the study, we set up a framework for automatic quantitative assessment of the conformance of the simulated ice area to the given geomorphologically determined ice bounds. This framework is novel in its ability to work with two types of bounds: clear and unclear, and evaluates the accuracy using two criteria. We also provide a simplification that combines these two criteria into one, which can then be used as an objective within the task of optimising computer model parameters.

line 65: "ARSO" has not yet been introduced at this point in the text.

"ARSO" is actually a misformatted reference (it is missing the year) at this particular location. We shall fix this reference in particular and also re-check the other references.

tables 1-3: I think they can be combined into one.

We considered joining the three tables into one but keeping the classification of parameters into three groups. The groups make sense as they introduce some order into an otherwise long unsorted list of parameters. However we see no gain in doing so, mostly because the third table is not compatible column-wise, since it presents different information.

figure 1: Why is the domain topography shown at a resolution of 25 m if the simulations use at most 50 m? Again, I would like to see the data on the original model grid without interpolation.

We tried to always show figures in the resolution that was used in the study. We never used any interpolation or other technique to artificially increase resolution of the figures. We have delved in more detail in our previous answer (see the answer on the first minor remark).

figure 2: The two shades of green are not optimal.

Indeed in the smaller of figures, the contrast is suboptimal. We are considering different contrast between the greens or turning light green into grey. We shall include an improved colour scheme in the final revision of the manuscript.

line 360ff: I am not sure this finding is very unexpected or important. Changes in temperature and precipitation may balance out and the effects are somewhat linear for small changes. This is how a Taylor expansion works.

We shall revise the paragraph by eliminating the first sentence and slightly modifying the second sentence to reference the figure. Indeed the "unexpected" part of the finding is more the fact that the relationship between temperature and precipitation can be expressed as round numbers that were chosen to be easily digestible, than a completely new finding.

figure 9: If I understood correctly, the red curve represents a simulation that stays at 400 m resolution even if the horizontal axis suggests otherwise. This is not very clear from either the legend of the figure caption and should be improved.

Yes, this is exactly what we are attempting to show on the figure but we are not conveying that information clearly. We shall modify the caption to include the following:

The label in the legend specifies the experiment's final resolution. Above the x axis, the grid

sequencing resolutions are specified, and times when resampling potentially occurs are marked with blue vertical dashed lines. Experiments are only resampled to higher resolutions at the resampling times if they have not reached the experiment's final resolution yet. For example, the 200 m experiment starts at 400 m resolution, refines to 200 m resolution at 1500 a but then remains at that resolution until 4000 a is reached and the simulation is stopped.

line 394: Is this result the same for all precipitation models? Which one was used here?

Yes, the result holds regardless of the precipitation model. The figure depicts results for simulation with the orographic model. We shall modify the Figure 10 caption:
Comparison of the simulation results that use the two presented temperature models and the orographic precipitation model.

We shall also modify the sentences that introduces Figure 10:

Figure 10 shows part of the sensitivity study for the insolation-adjusted temperature model. The results are not sensitive to the setting of the insolation effect amplitude. Only for values of 5 or greater, which seem unrealistic and are thus not shown on the figure, the resulting ice field shifts towards the north noticeably.

line 424: I think this relationship between T and P makes the original goal of reconstructing climate impossible without additional data that constraints either T or P. Most problems in the geosciences are underdetermined, but this one to a degree that needs to be addressed explicitly.

We believe that we are saying the same in the manuscript. We recognise that our wording need some improvement, thus we will change the paragraph to read:

In this subsection we present the results that to some extent address the goal of the study – improving our understanding of the past climate-glacier dynamics at the Alps-Dinarides junction. There are two main results. First, we establish that the problem is underdetermined and we can provide optimal climate conditions for formulation of ice field on plateau Snežnik only as a linear relation. Second, we find one set of climate conditions that respect both the linear relation from the first result, and the state-of-the-art global climate estimates. We present the modelled ice field under such climate conditions on the domain with resolution of 50 m.

line 475: I think calling the simulations "consistent with emperical geomorphological reconstructions" is a bit too overconfident. There are systematic biases.

We agree that the wording used here was misleading and we failed to acknowledge the problems of the proposed reconstruction. We will clarify our conclusions:

The presented relation between temperature and precipitation presents a degree of freedom that can only be resolved by additional external data. Latest reconstructions of LGM climate (Del Gobbo et al., 2023) are a great source for external data, and the proposed relation between temperature and precipitation matches this particular data point well. Simulations carried out under the climatic conditions of the LGM suggest an ice field that is broadly

consistent with empirical geomorphological reconstructions. The consistency is limited to ice field size however, as the simulations fail to reproduce all the bounds of the geomorphologically reconstructed ice field. Although the established framework aided in the optimization of the unbound parameters in climate models, some systematic biases remain in the simulations. These could be resolved in the future by using more detailed climate models. The precipitation model was a key component in the presented study and remains a candidate for further improvements.

line 484f: It should be noted that this relative insensitivity is found for the particular case at hand, not in general.

We shall rewrite the statement to emphasise that this finding is not general:

Conversely, the adequacy of elementary temperature models relying solely on lapse rates and elevation data reveals the relative insensitivity of the studied ice field extent to the temperature data.

line 487: "The study reveals that..." This statement should be removed. This fact is well known on a general level and the SMB model is not detailed enough to provide a deeper insight into the T/P compensation effect.

We will change this sentence in particular as copied below added some text (also copied below) and in combination with other changes in the manuscript we believe the tone regarding the T/P compensation will be adequately softened, with the emphasis shifted towards its quantification and determination of its limits within the study area:

The reconstructions resulted in the demonstration of the relation between air temperature and precipitation when it comes to the size of glaciers.

This study formulates the interplay as a pair of equations with a single independent variable. Furthermore it quantifies the parameters of equations and bounds them to a range of values beyond which the interplay gradually loses its effect.

line 493f: "This model is consistent with geomorphological field data..." Again, I think the simulation results are not good enough for that. The methodology allows for the quantification of the simulation skill, which is a step in the right direction. I do, however, agree with the second half of the same sentence, i.e., that it is "a convincing demonstration of the effectiveness of [..] integrating [..] models with [..] data".

We will soften the tone on the statement about consistency with the most relevant part now reading as follows:

Simulations carried out under the climatic conditions of the LGM suggest an ice field that is broadly consistent with empirical geomorphological reconstructions. The consistency is limited to ice field size however, as the simulations fail to reproduce all the bounds of the geomorphologically reconstructed ice field. Although the established framework aided in the

optimization of the unbound parameters in climate models, some systematic biases remain in the simulations.

Citation: https://doi.org/10.5194/egusphere-2024-544-RC1

The manuscript here presents the modelling of Snežnik, within the Dinaric Mountains in Slovenia, with the aim to present the likely climatic conditions glaciers existed under within the LGM. They use previously studied geomorphological evidence to determine, out of differing climate modification combinations, the 'best-fit model'. This study also takes a look into differing model inputs of climate forcing that can effect the model output.

Overall, the aim of determining the potential climate conditions is, in my opinion, not entirely reconciled in this study. This is due to there being poor constrains (i.e., no palaeorecords) on regional palaeotemperature, or palaeoprecipitation, presented by this study at the time of the LGM glaciation. The use of climatic offsets in the model do however, present a 'envelope' of climate conditions with differing precipitation and temperature offsets that, in combination, could generate ice at the geomorphologically constrained LGM extent.

Some major remarks from my review of the manuscript are:

1. As stated above, the determination of the climate conditions cannot rest on singular values of precipitation and temperature offsets. As mentioned in the text, "[Line 445] any increase in precipitation can be countered by a decrease in temperature to keep the conditions for simulated glacier formation about the same". A climatic envelope is more realistic as there is still a large amount of uncertainty in the ice reconstruction, the palaeoclimate, and within the model itself. Further, a climate envelope allows for there to be differing variations in the precipitation and temperatures, that may see palaeorecords fall within the stated range. Following on, a statement on what records could be used, or are needed, to allow a more constrained climate estimate would allow future studies to understand where the gaps are.

Hoping that we understand each other correctly, we would like to first reiterate that we provide two climate reconstruction results (we reorganised the conclusions to make this clearer). The one pointed out is a particular set of temperature and precipitation offsets (resulting in Fig 14), while the other is a relation between temperature and precipitation offsets Eq. (3). The first result is only one point within the range of possible results allowed by the linear relation. We single it out as the most feasible result, since it also matches a recent LGM reconstruction. We also experimentally determine a feasible range for the relation (Fig 15), which is a step towards creating an envelope.

Of course, on top of one degree of freedom that is given in our result, there come several more degrees of freedom originating in the uncertainty of the other model parameters. However, there is much more research needed in quantifying the uncertainty which would then lead to the ability to quantify the climate envelope. We do not think such an endeavour is even possible at this point.

Finally, if we may rephrase the question as "What knowledge would help the most in constraining the unknowns the best?" Our answer would then be the climate models. Not knowing the uncertainty in other model parameters of course makes this a very subjective claim. On the other hand, we have shown in the presented study that wind is a strong factor that can significantly alter the simulation and its effects are not uniform on the simulation domain (and is currently not even fully modelled, only its effect on precipitation are partly modelled). For example, nearly disjunct ice coverages can be achieved by just rotating

predominating winds between 120° and 240° angles (Fig 6). Therefore, since wind is a known important factor while others factors are unknown (although possibly also important), we feel it is only sensible to strongly prioritise wind.

We will be adding the following to discussion and conclusions:

As a phenomenon that is underrepresented in models but has a demonstrated high influence on results, wind should represent an area of additional research in the future.

2. While there has been an attempt to understand how the reconstructed ice is influenced by using differing climatic inputs (temperature forcing using lapse-rate model or WorldClim, precipitation forcing using single value or WorldClim), there is limited expansion on the model sensitivity. Certain model parameters will shift what climatic offsets are needed, and that may change certain numerical outcomes of the study. Within the PDD model for example, Degree Day Factors (DDFs) are likely to cause substantive differences in ice generation, as well as the refreezing factor. Other glaciological physical parameters (enhancement factors etc.), and till parameters (till water content etc.) may also effect the output when using the same temperature and precipitation offsets. I would expect there to have been some consideration of the model sensitivity to certain unconstrained parameters. If this has been done by previous studies within the same region, it needs to be stated within the text. I do understand that this follows off the back of Žebre et al. (2021) and Candas et al. (2020), that have done some sensitivity analysis previously, thus a more explicit indication of sensitivity of the model is needed.

The PDD factors were left at their defaults because of the lack of data that could be used to adapt them better for the study area (their values will be added to the table of parameters). Simulations were found to be very sensitive to them in previously published studies. The referenced study of Velebit is especially of value here, since the study area there neighbours the presented study area. We did perform some preliminary (i.e. not on the final set of models and parameters) sensitivity studies on several modelling parameters and obtained similar results as other studies did. Our baseline for sensitivity analysis was significantly different from the shown results (e.g. different precipitation/temperature ratio, different climate models and their parameters), thus we avoided adding specifics on the sensitivity analysis. However a near complete omission of discussing sensitivity is an oversight on or part and we shall add a paragraph to the manuscript, section "Conclusions" (which shall be renamed to "Discussion and conclusions"):

Another area where improvements should be sought is in optimising values of various unmentioned model parameters. Within the preliminary analysis of climate models we explored the sensitivity of the simulated ice field area and volume with varying modelling parameters such as those related to the domain grid, ice rheology, stress balance, basal sliding, and till properties. Our findings are consistent with those from studies published by Žebre et al. (2021) and Candaş et al. (2020). Specifically, the simulated glaciation extent and volume are as sensitive to choices related to parametrization of other models as they are to climate models. This sensitivity suggests that small variations in parameters can lead to significant differences in the results. Given this, along with the lack of local measurements to aid in parameter adjustments, the presented results should be interpreted with due caution. For future work, a methodology to set up or optimise all the major model parameters should be developed.

3. There is clearly a large amount of uncertainty in the geomorphological evidence, and the geochronology of the region. I did not see any statement on the timing of the LGM, nor

how it is known that these are LGM specifically? The studies cited to present the geochronology seemingly look at younger glaciation during the Younger Dryas. Žebre et al. (2019) does state that an age of 18.7 ± 1.0 cal ka BP was found from bone fragments in an outwash, but this is a singular piece evidence for the LGM. This does not definitively constrain the evidence as LGM specifically. The moraines used to constrain the model could be older/younger advances, which needs to have some appreciation in the introduction.

The statement about the timing is written in lines 60-61:

"The geochronological data (Marjanac et al., 2001; Žebre et al., 2019), **although still scarce**, points to a maximum ice extent during the last glacial maximum (LGM), that is 30–17 ka BP (Lambeck1 et al., 2014)."

We are honest about the uncertainty of the timing of moraines that have been used to constrain the model. We modified/added a sentence in the introduction (lines 25-27) to clarify the age uncertainty:

Snežnik was glaciated during the LGM, although the exact timing is still ambiguous (Marjanac et al., 2001; Žebre et al., 2016). Moraines that mark the farthest extent of the glacier have been attributed to the LGM, for which the maximum ice area was estimated to be at least 40 km2 (Žebre and Stepišnik, 2016).

4. Similarly to the above comment on the geomorphology used, the model reconstructs ice within the mountain range to the southeast (within the Gorski Kotar Ice Field), where no geomorphology has been presented (at least shown in this study). Žebre et al. (2016) does show (in their Figure 1) that ice was present on this high ground, with a question of ice filling a gap between two ice extensions (or where the ice is reconstructed in the model here). While it is understandable the domain is limited to the Snežnik icefield for computational reasons, a paragraph on the glaciation to the southeast would be beatifical, to present to the wider community where ice is being built, and where future studies should look for glacial evidence. This would also aid in locations which are loosely constrained (dashed lines in this study's Figure 1). Could the model aid in providing a likely area on where to look for evidence to more better constrain the LGM extent?

During geomorphological/geological field campaigns a much bigger area has been investigated from the one that has been limited to the maximum ice extent, but no glacial evidence has been found there. However, the model could potentially help finding areas hosting "hidden" glacial evidence, like for example deposits filling karst depressions.

An explanation about the glaciation of Gorski Kotar has been added to section 2.1:

Southeast of Snežnik lies the Gorski Kotar mountain range, which was also glaciated. Geomorphological mapping suggests that the Gorski Kotar ice field was approximately twice as large as that of Snežnik (Žebre and Stepišnik, 2016). Despite the close proximity of the two areas, there is no evidence to suggest that the two ice fields were connected (Žebre et al., 2016). Because of that and for computational reasons, the Gorski Kotar area was not included in our modelling domain.

5. Lastly, "[Line 471] This research has successfully established a quantitative framework for the assessment of palaeoglacial simulations that integrates both definitive and

provisional geomorphological deduced ice boundaries to improve the accuracy of model results". I agree this is one of the main features of the study to I believe warrants further consideration and development. Are there any major shortfalls that should be noted? How can this be improved upon for future uses?

We have added the shortcomings to the section "Discussion and conclusions":

The framework alleviated the difficult task of sorting simulation results by quality but did not eliminate visual checks entirely. The reason lies in its shortcomings, which we list here. First, its parameters require setup, which in turn requires experimenting. In the presented case the experimenting was light, but could have proven more difficult for a more demanding ice field shape. Second, the framework is missing a methodology for ignoring neighbouring glaciated areas. We expect neighbouring glaciations can often be a problem since the simulator requires the domain to be rectangular and to be somewhat larger than the area of interest. Thus, it is likely for most studies to find their domains contain some glaciers that are outside the focus and require special treatment in analysis. Finally, the proposed framework is very specific in its demands for at most two types of limits -- clear and unclear. There is currently no room for quantifying the clarity nor for including more limit types. While this framework would work if only clear limits were given, such cases could also make use of simpler analysis methods.

Below are minor remarks on the manuscript:

Line 2: State the study location name within the abstract within the location of the Dinaric Mountains.

We have introduced Snežnik in the first sentence of abstract:

In this paper we present a reconstruction of climate conditions during the Last Glacial Maximum on a karst plateau **Snežnik**, which lies in Dinaric Mountains (southern Slovenia) and bears evidence of glaciation.

Line 26: What is the timing of the LGM for this region?
Please see lines 60-61: "The geochronological data (Marjanac et al., 2001; Žebre et al., 2019), although still scarce, points to a maximum ice extent during the last glacial maximum (LGM), that is 30–17 ka BP (Lambeck et al., 2014)."
Line 36: 'extend' to 'extent'
We shall fix this typo.
Line 65: Citation issue I believe, ARSO needs a date.
Indeed, we will fix this in the bibtex as it appears that the chosen citation type does not support the date correctly.
Line 80: Superfluous information. Just state which version of PISM you used, and what it is briefly. Do not need to know what it was run on, unless on a HPC, then in acknowledgments state the HPC and ownership.
Agreed, this part of the information is irrelevant and will be removed (it was a local computer workstation).
Line 81: The study uses the same parameters are another study with the same author, how far away is the Žebre et al. (2021) study compared to this study?
The two study areas are only about 100 km distant from each other. We shall add this information to the manuscript too:
".. where a larger mountain range about 100 km SE of Snežnik…"
Line 83: I am surprised that 'most parameters were .. left at their default values.' A table

similar to that sin Candas et al. (2020) would be good, but I do understand there are a lot of tables that could also be combined.

This statement of ours originates in the fact that PISM offers hundreds of modelling parameters. For example, there are at least 28 parameters exposed for the PDD model alone. Their default settings are of course very sensible with the origin of the selected value cited in literature. However, to avoid confusion we shall simplify this part of the text to:

In this section, we list all the parameters that require explicit setting along with some of those that were left at their default values but were being analysed in preliminary testing and sensitivity analysis.

We have seriously considered merging the tables but have so far decided against it, since the current grouping of parameters improves the readability and since Table 3 is not really compatible (different columns) with the other two.

Line 84: 'e.i.,' to 'i.e.'

We will fix this typo.

Table 1: You use the -surface_pdd in PISM, but what DDF values were used?

We shall add the factors to the table 2
- Degree-day factor for ice 0.00879121 (meter / (Kelvin day))  -surface.pdd.factor_ice 0.00879121
- Degree-day factor for snow 0.0032967  -surface.pdd.factor_snow 0.0032967
- Refreeze 0.6   -surface.debm_simple.refreeze_ice_melt 0.6

Table 2: The wind direction (150°) is not inline with what is stated in the 'PISM option'

We added a note to explain the discrepancy:

Wind direction in PISM seems to ignore the coordinates supplied with DEM and instead assumes some default orientation of the supplied data. We supplied the data oriented differently, therefore the wind direction had to be remapped.

Line 89: 'maps' to 'models' as DEM stands for 'digital elevation model'

We will fix this oversight on our part.

Line 137: 'illustration' to 'illustrate'

We will fix the typo.

Figure 2: Placing the lines of the geomorphology would help the reader understand why certain regions are forbidden and others are not.

Indeed, since this figure serves to introduce the validation method, the lines are required and we will add them.

Line 189: I do not understand the sentence ...'precipitation model output is multiplied by a factor to either increase or decrease the precipitation linearly by several percent to several ten percent'. However later you say on line 357 "...with air temperature spacing by 0.5 °C and precipitation spacing of 10% are shown". Is this a different test or climate offset you are using from that stated before, or the same? Having a varying percentage difference for precipitation makes it confusing to know what percentage change you use for which precipitation offset.

We agree that the sentence in question is rather confusing and we are modifying it as follows:

… the precipitation model output is multiplied by a factor to cause a relative reduction or amplification, which is then expressed in percentages.

Figure 3: Figure caption require more information. Where is the data from? Elevation of the AWS? What is the period this is for?

While all the data is in the text that introduces the figure, we shall add the relevant numbers also to the caption to make the figure self-contained.

Figure 5: Put what the precipitation model is above the figure boxs, similar to Figure 4.

We shall fix the figure, the model names should definitely be there.

Line 244: What interpolation technique was used?

Lanczos resampling was used. We will clarify it in the text. We will also fix the error in software mentioned, we actually bypassed QGIS and used GDAL directly:

We use the WorldClim (Fick and Hijmans, 2017) model of global climate as the source, reduce its coverage to the area of the domain and interpolate (with Lanczos resampling) the mean monthly precipitation component on a 50 m resolution grid (all of the above was done in software package GDAL).

Line 261: Same as above

This text is about the same procedure as the one in remark above. It is not a separate interpolation.

Figure 6: Second line down 'hte' ???

We will fix the typo.

4 Results: If there is no discussion section, name this as 'Results and Discussion'.

Agreed, the discussion section is missing. However, we have added the discussion to conclusions and renamed that section appropriately, since it already contained some of the discussion. There have also been several other discussion paragraphs added to that section.

Line 357: ...'air temperature spacing by 0.5°C and precipitation spacing of 10%' - The spacing of temperature is correct. But the spacing of the precipitation is not on Figure 8. You say it is a 10% spacing from the 2041 mm baseline. If this was correct, you would be adding 204.1 each time. However, between 2041 and 2320 is 279 (13.7%), while between 2320 (the first column) and 2610 (the second column) is 290 being 14.2% of 2041 or 12.5% of 2320. If the starting point is correct, please correct what you actually used as a percentage spacing for your precipitation offsets.

There is an error on our side -- we had found an error in the simulations for the figure (these were done very early on in the study) and we have redone the simulations using very different settings, including a different central point. The 0.5°C and 10% steps in the matrix are correct, but the starting point was changed to -7.5°C and 2900 mm precipitation. The offending line was removed from the manuscript (correcting it would still be misleading without adding to the content).

Line 358: You say you start the temperature and precipitation offset at -6°C and 2041 mm/a respectively, but do not show it on Figure 8. If you do not show it, state the values as the first model run shown with the lowest numbers used.

In line with the previous response, this was an error and is removed now.

Figure 8: Due to how far the figure is from the initial figure on the colours and their meaning, place them on the figure so we know what they mean. Further, you use °C and K interchangeably, stick with one or the other.

We have changed Kelvins to °C in several figures.

Line 360: 'An decrease in temperature of 0.5°K coupled with a 10% decrease in precipitation does not significantly alter the extent of ice field.' - I think there is something wrong here, and that you mean, a increase of 0.5°C, coupled with a 10% decrease in precipitation, or vice versa, does not significantly alter the ice field extent.

No, the sentence is actually correct (although the units will be changed to °C). One needs to lower temperatures to balance out a decrease in precipitation. The confusion arises because the temperature offset is negative from the start, and it needs to be more negative (which makes its magnitude larger) to balance out a decrease in precipitation.

Figure 9: While I think I know what is going on, it is not well explained. From the figure, each line colour represents a resolution, is it that they were not resampled past their resolution? So they all start at 400m, and the redline is if it was never resampled, while the green line is when it is resampled to 200 m at 1500 and then not beyond that (i.e., not resampled to 100 m at 2500)? Needs to be better explained.
Correct, this is exactly what we are attempting to show on the figure but we are not conveying that information clearly. We shall modify the caption to include the following:
The label in the legend specifies the experiment's final resolution. Above the x axis, the grid sequencing resolutions are specified, and times when resampling potentially occurs are marked with blue vertical dashed lines. Experiments are only resampled to higher resolutions at the resampling times if they have not reached the experiment's final resolution yet. For example, the 200 m experiment starts at 400 m resolution, refines to 200 m resolution at 1500 a but then remains at that resolution until 4000 a is reached and the simulation is stopped.

Line 413: "The larger ice field located to the southeast is partially responsible, which covers the largest part of forbidden area in the simulation" - Is this forbidden because it is known there was no ice there? Žebre et al. (2019) shows ice was in the Gorski Kotar Ive Field. If it is of a similar elevation and close to the study area here, why would it not have had ice during the LGM?
Since the part of Gorski Kotar is within the simulation domain, and its elevation is sufficient, some ice forms there as it should. Yet, the extent of ice from Gorski Kotar should not be trusted, because its accumulation zone is only partly included in the domain. Furthermore, it forms within the edge of the domain, exposing it to the influence of unrealistic border conditions. We wish to exclude it from the analysis in order to focus on what is believed to be a separate Snežnik ice field. There are multiple ways of excluding it, all with its own flaws.
The ways of excluding are out of scope for the manuscript, so we only explain it here:
One way would be to "fix" the topology - lower the Gorski Kotar so it does not produce the ice and thus not influence the research. Or modify the climate models in the same area to turn the climate unfit for ice accumulation.
Another would be to fix the results (remove that part of the ice) so that the analysis does not have to deal with them. The former ice fields of Snežnik and Gorski Kotar are however geographically so close together they often merge in simulations which could cause significant errors if one were artificially limited.
We opted for the third option which is acknowledging the ice field to the south-east but trying to ignore it in visual analysis and allowing it to have some limited influence on the automated analysis. After all, the area of influence is very limited and the proposed methodology should be robust to touch conditions to have any hope of being useful in general.
We shall include the following modification to the manuscript (in addition to acknowledging Gorski Kotar already in section 2.1):
The ice field to southeast is a part of otherwise known Gorski Kotar (Žebre and Stepišnik, 2016), and is expected to glaciate in simulations, but was not a part of the performed study. The isolation of the two ice fields is clear from the performed geomorphology studies. Its proximity, however, makes the analysis of Snežnik in isolation more demanding.

Line 422: Rather then the overall 'climate conditions' I believe that it is more of a climatic envelope under which these glaciers can exist under in this region. As there is no direct

control over the temperature or precipitation in the region from the palaeorecord, it is impossible to definitively determine the climate conditions needed.

Since this remark seems in line with the major remark 1, we refer to our answer there. In summary, the linear relation is not a singular result and does allow for very different precipitation and temperature conditions. Additional bounds to the temperature and precipitation are at this point not feasible since uncertainties in other model parameters cannot be quantified.

Figure 14: Addition of a colour ramp for the ice thickness, as elevation for the DEM is the only one shown.

We have alternatively chosen to unify the figure style with the other figures depicting ice thickness without the underlying topology, which increases the figure legibility. We hope this approach is also satisfactory.

Line 474: ...'produced an ice field that is consistent with empirical geomorphological reconstructions.' - While the 'optimal' simulation is closest to the geomorphological evidence, I do not think it can be said that it is consistent with the evidence, as there are areas where it is not. Maybe saying that it falls the closest to the geomorphological evidence, and maybe state where there are still regions that cannot be reconciled here?

We agree that the wording used here was misleading and we failed to acknowledge the problems of the proposed reconstruction. We will clarify our conclusions, starting with the sentence in line 474:

The presented relation between temperature and precipitation presents a degree of freedom that can only be resolved by additional external data. Latest reconstructions of LGM climate (Del Gobbo et al., 2023) are a great source for external data, and the proposed relation between temperature and precipitation matches this particular data point well. Simulations carried out under the climatic conditions of the LGM suggest an ice field that is broadly consistent with empirical geomorphological reconstructions. The consistency is limited however, as the simulations fail to reproduce all the bounds of the geomorphologically reconstructed ice field. Although the established framework aided in the optimization of the unbound parameters to climate models, some systematic biases remain in the simulations. These could be resolved in the future by using more detailed climate models. The precipitation model was a key component in the presented study and remains a candidate for further improvements.

Figure 16: Seems there is the southeastern region that does not get entirely covered by the model ever - would be good to see a small sentence or section just recognising it and musing why this may be.

Since not all the performed simulations are shown in results, the article might be a bit misleading in this regard, however showing everything is also not feasibly. In reality, all parts of the ice field can be simulated, given enough tweaking of the climate models. However, we were never able to simulate them all within the same simulation run. The presented results show only some of the best trade-offs between overrepresentation on one, and underrepresentation on another section of the prescribed ice field. For example, the eastern-most part of the glacier is not in any of the shown results but can become even over-expressed relative to other parts of the ice field if different wind direction is chosen to govern the orographic precipitation.

Lines 487: 'The study reveals that air temperature and precipitation are closely linked when it comes to the size of glaciers.' - This is a well known relationship in models, I do not think it is really the most important result from this study.

We have changed the first sentence and emphasised more that this study quantifies the

effect and puts the limits on its reach:

The reconstructions resulted in the demonstration of the relation between air temperature and precipitation when it comes to the size of glaciers. These two factors affect each other in a way that creates a balance within the simulations causing different combinations of temperature and precipitation to result in similar glacier extent. This interplay precludes the determination of exact climatic conditions based on glacier morphology alone and suggests a broader framework of possible past climates that are consistent with the observed glacier extent. This study formulates the interplay as a pair of equations with a single independent variable. Furthermore it quantifies the parameters of equations and bounds them to a range of values beyond which the interplay gradually loses its effect.

Line 495: Potentially provide how the model shows uncertainty on some of the geomorphological evidence. The model could be used to provide areas that need to be looked at in further detail.

The areas with the least geomorphological evidence (unclear outline) have been researched in the field several times. So far, no geomorphological evidence in the form of marginal glacial landforms/sediments has been found in these areas. More evidence for glaciation could be potentially found by drilling into the sediments trapped in karst depressions close to the simulated ice boundary.

We have added to conclusions:

On the other hand, the existing simulations can be used to find additional evidence of glaciation in the field to better constrain the next iteration of the simulation. While a larger area of Snežnik has already been mapped and every standard-sized landform has been mapped, there is still much that is unknown when it comes to the sediment filling of karst depressions. The existing best-fit simulations could potentially help locate points of interest for further research, e.g. to plan drilling into the sediments trapped in karst depressions close to the simulated ice boundary.

I have attached a PDF that has the locations of these minor remark to aid in editing.

Overall, I believe the work can use some tidying up, with further detail on, 1) certain sources of information that the study is resting itself on (geochronology, geomorphology etc.), 2) the model sensitivity, as this can substantially change the interpretation if different values are used for certain unconstrained parameters, and 3) that some form of track change is needed to present these as an envelope of climate combinations that allow ice to sit at the likely LGM extent here. I look forward to seeing more research from these authors in the future.

Citation: https://doi.org/10.5194/egusphere-2024-544-RC2